# Intra-crystalline mesoporous zeolite encapsulation-derived thermally robust metal nanocatalyst in deep oxidation of light alkanes

Honggen Peng [1,10 ✉], Tao Dong[1,10], Shenyou Yang[1], Hao Chen[2,3], Zhenzhen Yang[2,3], Wenming Liu[1], Chi He [4 ✉], Peng Wu [5], Jinshu Tian[6], Yue Peng [7], Xuefeng Chu[7], Daishe Wu[1], Taicheng An [8 ✉], Yong Wang[9] & Sheng Dai [2,3 ✉]

Zeolite-confined metal nanoparticles (NPs) have attracted much attention owing to their superior sintering resistance and broad applications for thermal and environmental catalytic reactions. However, the pore size of the conventional zeolites is usually below 2 nm, and reactants are easily blocked to access the active sites. Herein, a facile in situ mesoporogen-free strategy is developed to design and synthesize palladium (Pd) NPs enveloped in a single-crystalline zeolite (silicalite-1, S-1) with intra-mesopores (termed Pd@IM-S-1). Pd@IM-S-1 exhibited remarkable light alkanes deep oxidation performances, and it should be attributed to the confinement and guarding effect of the zeolite shell and the improvement in mass-transfer efficiency and active metal sites accessibility. The Pd—PdO interfaces as a new active site can provide active oxygen species to the first C—H cleavage of light alkanes. This work exemplifies a promising strategy to design other high-performance intra-crystalline meso-porous zeolite-confined metal/metal oxide catalysts for high-temperature industrial thermal catalysis.

[1] Key Laboratory of Poyang Lake Environment and Resource Utilization, Ministry of Education, School of Resources Environmental and Chemical Engineering, College of Chemistry, Nanchang University, 999 Xuefu Road, Nanchang, Jiangxi 330031, China. [2] Chemical Sciences Division, Oak Ridge National Laboratory, Oak Ridge, TN 37830, USA. [3] Department of Chemistry, University of Tennessee, Knoxville, TN 37996, USA. [4] State Key Laboratory of Multiphase Flow in Power Engineering, School of Energy and Power Engineering, Xi'an Jiaotong University, Xi'an 710049 Shaanxi, China. [5] Shanghai Key Laboratory of Green Chemistry and Chemical Processes, Department of Chemistry and Molecular Engineering, East China Normal University, North Zhongshan Road 3663, Shanghai 200062, China. [6] Technology and College of Chemical Engineering, Zhejiang University of Technology, Hangzhou 310014, China. [7] State Key Joint Laboratory of Environment Simulation and Pollution Control, National Engineering Laboratory for Multi Flue Gas Pollution Control Technology and Equipment, School of Environment, Tsinghua University, Beijing 100084, China. [8] Guangdong Key Laboratory of Environmental Catalysis and Health Risk Control, School of Environmental Science and Engineering, Institute of Environmental Health and Pollution Control, Guangdong University of Technology, Guangzhou 510006, China. [9] The Gene and Linda Voiland School of Chemical Engineering and Bioengineering, Washington State University, Pullman, WA 99164, USA. [10]These authors contributed equally: Honggen Peng, Tao Dong. ✉email: penghonggen@ncu.edu.cn; chi_he@xjtu.edu.cn; antc99@gdut.edu.cn; dais@ornl.gov

Volatile organic compounds (VOCs) are the major air pollutants emitted from the stationary and mobile combustion of fossil fuels[1–3]. VOCs not only participate in the formation of toxic ozone with nitrogen oxides ($NO_x$) and photochemical smog but also are the main category greenhouse gases that lead to global warming[1]. Recently, the release of light alkanes, typical VOCs, has attracted much attention[4–7] due to the strong C–H chemical bond in light alkanes being hard to degrade[8–10]. Therefore, the control of light alkane emissions is still a challenge. Deep oxidation by catalysts is believed to be one of the most efficient technologies to remove the light alkanes due to its moderate operating temperature and high removal efficiency and, importantly, it does not produce secondary pollution[11,12].

Generally, noble metal-based materials are considered one of the most efficient catalysts for deep oxidation of VOCs[1,2] because of their high activity and stability[13,14]. However, traditional supported noble metal-based catalysts are unsatisfactory for catalytic applications because particle aggregation occurs at high thermal reaction temperatures as a result of the Ostwald ripening process[15]. Recently, a new method has been developed to encapsulate the precious metal NPs within crystal zeolites, which can greatly hinder the aggregation of precious metal NPs[16–20]. Nevertheless, there are still existing some problems for these microporous zeolites confined catalysts because the pore size of the conventional zeolites is usually below 2 nm and limits the molecule diffusion and the metal active sites accessibility[21,22]. Therefore, promoting the mass-transfer efficiency of reactants and products and accessibility of the active sites over the conventional zeolite confined catalysts is an urgent problem for various catalytic reactions, e.g., the deep oxidation of VOCs. A potential good strategy to resolve this problem is to synthesize a mesoporous-zeolite shell to promote the mass-transfer efficiency and accessibility of the active sites[23–26]. Though many strategies (demetallization method, soft- and hard- template methods, etc.) have been developed to obtain mesoporous zeolites[27–31], to the best of our knowledge, it is still a challenge to fabricate single-crystal zeolites with intra-crystalline mesopores (intra-mesopores) and confine the active metal NPs simultaneously via a simple in situ mesoporogen-free strategy. Because it is still difficult to obtain mesoporous zeolites, and some disadvantages are not overcome such as high costs, health hazards, limitations on the types of synthetic zeolites, and decreased the crystallinity of the zeolites.

Herein, the single-crystalline zeolite silicalite-1 (S-1) with intra-mesopores encapsulating palladium (Pd) NPs (Pd@IM-S-1) was successfully synthesized through a facile in situ mesoporogen-free strategy. This work was different from the previous works which are only containing the single microporous channels, and this work contains micropores and the supernumerary intra-mesopores simultaneously, and the synthetic method was compared with the previous works as shown in Supplementary Fig. 1 Transmission electron microscopy (TEM), high resolution TEM (HRTEM), and focused ion beam (FIB) aberration-corrected TEM (FIB-AC-TEM) strongly verified that abundant mesopores exist in the nanocrystals and Pd NPs are confined in the zeolite shell. To clarify the formation mechanism of intra-mesopores of Pd@IM-S-1, the crystallinity and morphological evolution with different crystallization time have been investigated by X-ray diffraction (XRD) and TEM characterization methods. Meanwhile, the universality of the facile in situ mesoporogen-free method was also investigated, and results show that the facile method exhibits versatility and potential applicability. Furthermore, the typical light alkanes methane and propane were selected as the model compounds to investigate the deep oxidation performance of Pd@IM-S-1. To our delight, compared with the supported catalyst (Pd/S-1), Pd@IM-S-1 exhibited remarkable methane and propane deep oxidation activities, thermal and water resistance. The density functional theory (DFT) calculations, the in situ diffuse reflectance infrared Fourier-transform spectroscopy (in situ DRIFTS), and the in situ near ambient pressure X-ray photoelectron spectroscopy (In situ NAP-XPS) characterization results demonstrate that the Pd—PdO interfaces can provide the active lattice oxygen species ($O^-$) to oxidation of light alkanes.

## Results

**Synthesis and physicochemical properties of Pd@IM-S-1.** As illustrated in Fig. 1, Pd@IM-S-1 was prepared via a facile two-step, mesoporogen-free in situ method using amorphous porous silica-confined Pd NPs ($Pd@SiO_2$) as the precursor. TEM and HRTEM images of $Pd@SiO_2$ are shown in Supplementary Fig. 2. It is evident that multiple amorphous silica-confined Pd NPs (~1.3 nm) were successfully synthesized. Subsequently, Pd@IM-S-1 was obtained from $Pd@SiO_2$ as the precursor through an in situ dry-gel mesoporogen-free conversion process. The powder XRD patterns demonstrate that Pd@IM-S-1 displays the five typical diffraction

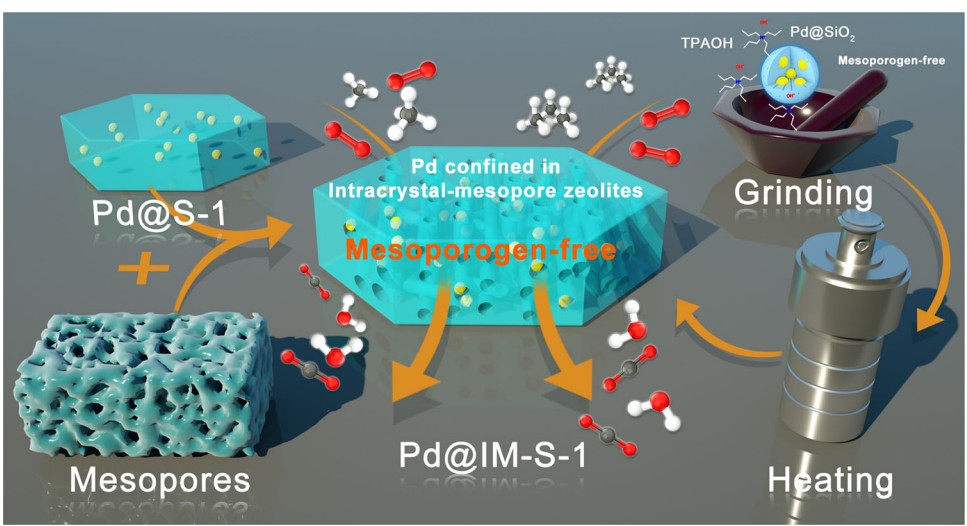

**Fig. 1 Schematic illustration for the synthesis of Pd@IM-S-1 and application for deep oxidation of light alkanes.** Pd@IM-S-1 was synthesized via the two-step mesoporogen-free in situ method.

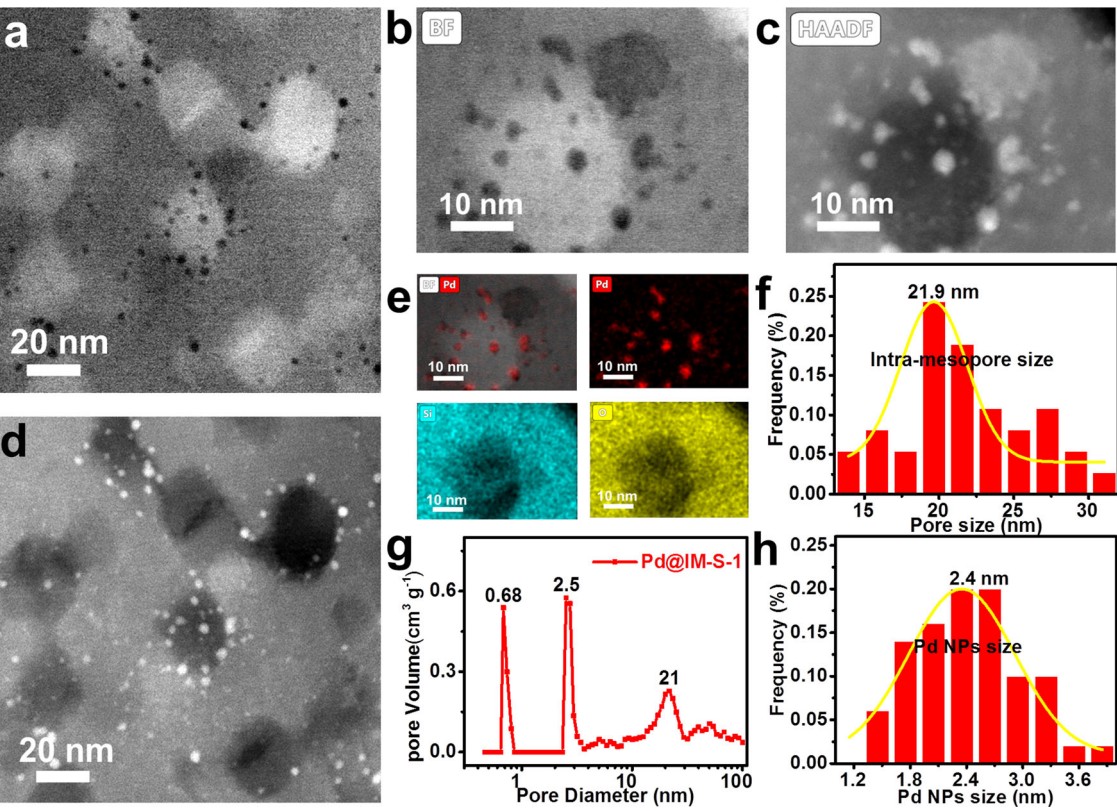

**Fig. 2 Morphology and structural characterization of Pd@IM-S-1. a**, **b** FIB-AC-BF-TEM, **c**, **d** FIB-AC-HAADF-TEM, **e** FIB-AC-EDS-mapping, **f** intra-mesopore size distribution, **g** pore size distribution curve, **h** Pd NPs size distribution.

peaks of a typical MFI crystalline structure (Supplementary Fig. 3). The amorphous silica phase was not observed, unlike its precursor (Pd@SiO₂), which means that the amorphous silica shell was transformed into a highly crystalline zeolite structure shell and is similar to the pure silica zeolite S-1 and Pd/S-1 prepared by the conventional impregnation method. The inductively coupled plasma optical emission spectrometry (ICP-OES) analysis confirmed that the Pd content of Pd@IM-S-1 was about 1.8 wt.% as listed in Supplementary Table 1.

In the TEM images of Pd@IM-S-1 in Supplementary Fig. 4a, b, it is evident that abundant intra-mesopores are present in the nanocrystals. Interestingly, the SEM and HRTEM images (Supplementary Figs. 5 and 6a) and the selected area electron diffraction images (Supplementary Fig. 6b) reveal that the whole nanoparticle is similar to a single crystal, even with enriched mesopores inside the entire particle[32], and the Pd NPs are encapsulated within the single-crystalline mesoporous zeolite shell. The HAADF-STEM and elemental energy-dispersive X-ray spectroscopy (EDS)-mapping images and line scan profiles of Pd@IM-S-1 are shown in Supplementary Fig. 7a–f. The elemental EDS-mapping images evidently demonstrate that the Pd element is homogeneously distributed over all of the zeolite. And the line scanning profile (Supplementary Fig. 7f) demonstrates that the Pd species are confined in the single-crystalline mesoporous S-1 shell, which is confirmed by the TEM images. In contrast, the Pd NPs over Pd/S-1 prepared by a conventional impregnation method are unevenly dispersed on the outside of the S-1 zeolite (Supplementary Fig. 8a, b).

In order to further confirm that Pd@IM-S-1 possesses abundant intra-mesopores and Pd NPs are indeed confined within the mesopores of the zeolite shell, the FIB technique was performed to prepare specimens for aberration-corrected TEM characterization. The detailed preparation process and simple

flowchart (Supplementary Fig. 9) of FIB specimens are presented in the Supplementary Information. The FIB aberration-corrected bright field TEM (FIB-AC-BF-TEM, Fig. 2a, b) and FIB aberration-corrected high-angle annular dark-field TEM (FIB-AC-HAADF-TEM, Fig. 2c, d) images of Pd@IM-S-1 clearly show that a large number of intra-mesopores are present in the nanocrystals and Pd NPs are confined within the IM-S-1 shell. The mean size of the observed intra-mesopores, 21.9 nm (Fig. 2f), can enhance transportation of the reactants and products and accessibility to the Pd active centers. The mean size of tiny Pd NPs was 2.4 nm (Fig. 2h), and the mean size of larger Pd NPs was 4.0 nm (Supplementary Fig. 10a) and the largest Pd NPs was up to 13.2 nm or 18.4 nm (Supplementary Fig. 10c). The results indicate that though some Pd species are aggregated into large particles, most of the Pd species still exist in the form of small particles. In addition, the FIB aberration-corrected EDS-mapping images (FIB-AC-EDS-mapping, Fig. 2e and Supplementary Fig. 10d−f) also proved that Pd species are homogeneously distributed within the mesoporous zeolite shell.

The nitrogen physisorption curves of Pd@IM-S-1, S-1, Pd/S-1, and Pd@SiO₂ are shown in Supplementary Fig. 11 and the detailed results are listed in Supplementary Table 1. As shown in Supplementary Fig. 11b, Pd@IM-S-1 displays two types of curves with three hysteresis loops at a relative pressure of < 0.05, 0.2–0.45, and 0.45–1.0, indicating that a trimodal porous structure has been formed[23,26,33]. Typically, the loop in the relative pressure $P/P_0 < 0.05$ can be attributed to the micropores of the S-1 shell (0.68 nm, calculated by the Horvath-Kawazoe method) (Fig. 2g), and the loops in the relative pressure regions of 0.2–0.45 and 0.45–0.85 are due to the intra-mesopores derived from the zeolite shell. The dimension size of the two kinds of mesopores was 2.5 and 21 nm (Fig. 2g), respectively, derived from the Barret−Joyner−Halenda method. The large mesopore size

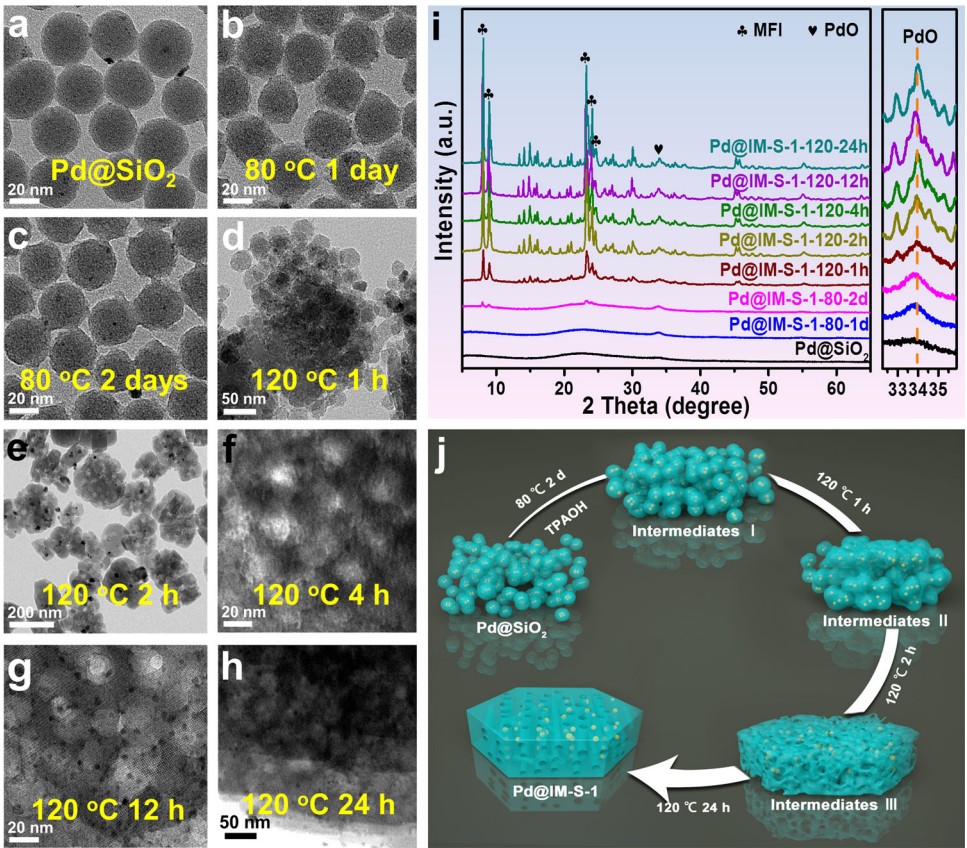

**Fig. 3 Formation process of Pd@IM-S-1. a** TEM image of Pd@SiO$_2$, **b**, **c** TEM images of Pd@IM-S-1 crystallized 1 day and 2 days at 80 °C, respectively; **d–h** TEM images of Pd@IM-S-1 crystallized 1 h, 2 h, 4 h, 12 h, and 24 h at 120 °C, respectively; **i** XRD patterns of Pd@IM-S-1 crystallized between 80 and 120 °C at different crystallization time; **j** proposed formation process of intra-mesopore over Pd@IM-S-1.

(21 nm) is well in line with the pore sizes observed from the FIB-AC-TEM images (21.9 nm). Furthermore, the specific surface area of Pd@IM-S-1 was sharply increased to 408 m$^2$ g$^{-1}$, while that of its precursor (Pd@SiO$_2$) was only 93 m$^2$ g$^{-1}$ (Supplementary Table 1). The larger surface area facilitates the exposure of metal active sites to enhance the catalytic performance. Therefore, Pd@IM-S-1 was successfully prepared by a facile two-step mesoporogen-free strategy without the involvement of an additional mesopore-creating agent. The introduction of mesopores exposes more accessible active centers and is beneficial to the diffusion of the reactants and products.

To understand the formation mechanism of Pd@IM-S-1, the crystallinity and morphological evolution at different crystallization time have been monitored by XRD and TEM. During this test, all samples are unreduced, and the detailed experimental process is presented in the Supplementary Information. As shown in Fig. 3i, the Pd@IM-S-1-80-1d still maintains the amorphous SiO$_2$ after being aged at 80 °C for one day. The TEM images of Pd@IM-S-1-80-1d proved that partial dissolution of the SiO$_2$ shell occurred (Fig. 3b). The Pd@IM-S-1-80-2d sample detected new diffraction peaks ascribed to MFI crystalline structure when crystallization time increased to two days at 80 °C (Fig. 3i). The corresponding TEM images (Fig. 3c) show further dissolution of the shell SiO$_2$, and spherical particles are beginning to connect to each other. The results of Fig. 3i and 3c indicated that the interfaces of amorphous silica-zeolite are formed. Subsequently, the crystallinity of Pd@IM-S-1 gradually increased from 1 to 2 h at 120 °C, and the crystallinity of Pd@IM-S-1 remains unchanged after the crystallization time further increased to 4 h at 120 °C (Fig. 3i). It is observed in Fig. 3d that the multiple small Pd@SiO$_2$ particles are connected more tightly and begin to reorganize into

large Pd@IM-S-1 particles. The morphology is further evolved, and some mesopores are observed over Pd@IM-S-1-120-2h as shown in Fig. 3e. From Fig. 3f–h, it could be observed that the quantities of mesopores are increased evidently, while the morphology remains unchanged.

Based on the results mentioned above, it can be concluded that the growth of Pd@IM-S-1 is a fast dissolution and recrystallization process, and the formation process of Pd@IM-S-1 is shown in Fig. 3j. The microporous structure-directing agent (tetrapropylammonium hydroxide, TPAOH) acting as a bifunctional reagent hydrolyzed into OH$^-$ and TPA$^+$, which etched the amorphous silica from Pd@SiO$_2$ and directed the formation of the zeolite framework. First, the silica shell of Pd@SiO$_2$ partially dissolved, and small particles are aggregated into bulk particles. The interstitial pores between particles appeared, due to the spherical materials close contacted with each other is impossible. Subsequently, the amorphous silica−zeolite interfaces are formed with the passage of crystallization time and the crystallinity of nanocrystals are further improved. In this process, the Si species and Pd species migrated and diffused among the amorphous silica−zeolite interfaces. However, the diffusion rates of the Si species and Pd species near the interfaces were rapid, whereas the Si and Pd species far away from the interfaces were slow to diffuse, due to the difference in diffusion rates and the presence of interstitial pores. Finally, generating mesopores and confining the Pd species in the mesopores occurred simultaneously. A similar growth mechanism was also found by other groups[26,33].

Moreover, the universality of this synthetic method for the preparation of other noble metals confined within the IM-S-1 zeolite shell in situ was also investigated. Typically, the IM-S-1, 1%-Pt@IM-S-1, 1%-Rh@IM-S-1, and 1%-Ru@IM-S-1 are

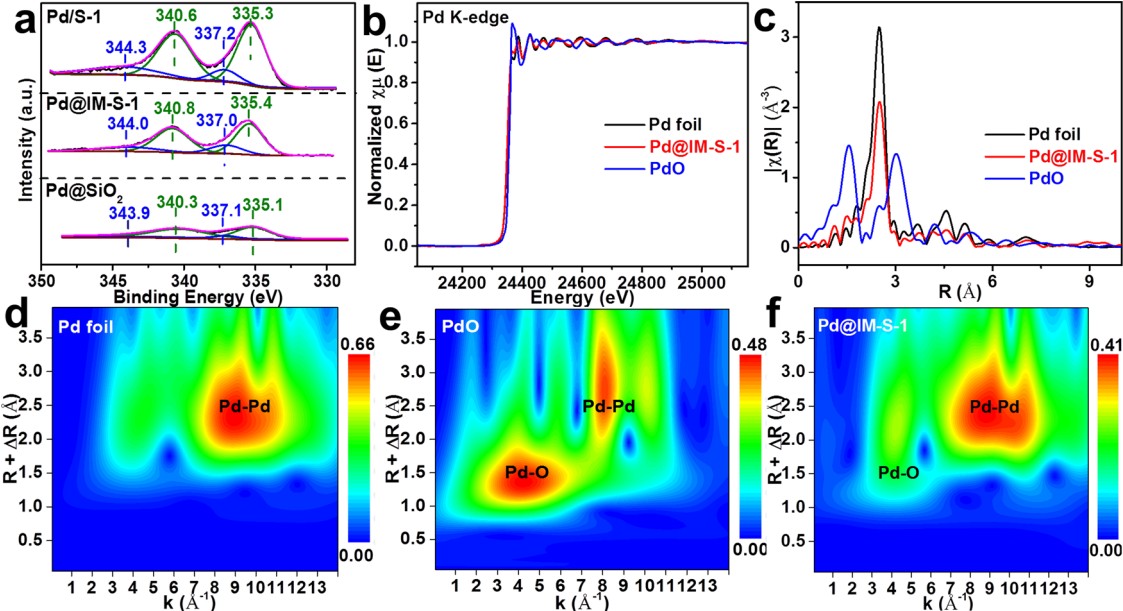

**Fig. 4 Valence state and Pd-PdO interaction of Pd@IM-S-1. a** XPS of Pd3d for Pd@IM-S-1, Pd/S-1 and Pd@SiO₂; **b** XANES spectra at the Pd K-edge; **c** Fourier transform of k3-weighted EXAFS spectra at the Pd K-edge; **d−f** the wavelet transforms from experimental data for Pd foil, PdO, and Pd@IM-S-1.

prepared via the facile in situ mesoporogen-free two-step method. The XRD patterns proved that high crystallinity of MFI type zeolite structure was successfully obtained as shown in Supplementary Figs. 12, 14, 17, and 20. The TEM images (Supplementary Figs. 13, 15, 18, and 21) showed that abundant intra-mesopores are present in these nanocrystals, and most of Pt, Rh, and Ru NPs are confined in the IM-S-1 zeolite shell, respectively. Furthermore, the Tomogram-section TEM was performed to further prove the Pt, Rh, Ru are confined within IM-S-1 zeolite shell. As shown in Supplementary Figs. 16, 19, and 22, most of the Pt, Rh, Ru are indeed confined within IM-S-1 zeolite shell and the mean sizes of Pt, Rh, Ru NPs are 5.8, 2.3, 2.3 nm, respectively. The above results show that the facile method exhibits versatility and potential applicability and can be used to confine other noble metals or metal oxides in the intra-mesopores zeolite shell.

To determine the status of Pd, Raman spectra, XPS, X-ray absorption near-edge structure (XANES), and the extended X-ray absorption fine structure (EXAFS) were acquired for Pd@IM-S-1 and related catalysts. Above all, the Raman spectra of the unreduced catalysts are displayed in Supplementary Fig. 23a; the peaks at 277 and 343 cm$^{-1}$ might be ascribed to the transverse acoustic mode of Raman spectra[34]. Bands at 441 and 653 cm$^{-1}$ are assigned to the $E_g$ and $B_{1g}$ of the Pd–O bond[35,36]. Then, the reduced Pd@IM-S-1 and related catalysts were also characterized by the Raman technique (Supplementary Fig. 23b). It is evident that the intensity of the Raman peaks around the Pd–O bond over Pd@IM-S-1 can rarely be observed, and the intensity over Pd/S-1 is decreased—only a weak peak can be observed. The Raman results initially indicated that most of the PdO species were reduced to metallic Pd species, but some PdO species were still unreduced because of the strong interaction between the Pd–PdO interfaces[37].

The XPS spectra of the reduced Pd@IM-S-1, Pd/S-1, and Pd@SiO₂ were acquired to further determine the status of the Pd species; the results are displayed in Fig. 4a. The Pd species of the three catalysts have two valence states—Pd$^{2+}$ and Pd$^0$. Typically, the Pd $3d_{3/2}$ peaks at 343.9–344.3 eV and 340.3–340.8 eV are attributed to the Pd$^{2+}$ and Pd$^0$ species, respectively, as are the Pd $3d_{5/2}$ peaks at 337.0–337.2 eV and 335.1–335.4 eV[38–42]. The XPS results also confirmed that some PdO species are still unreduced.

As shown in Fig. 4b, the Pd K-edge XANES spectra of Pd@IM-S-1 exhibited some differences from PdO reference spectra but similar to Pd foil reference spectra. The results indicated that the metal Pd species (Pd$^0$) dominates the Pd species status of the Pd@IM-S-1, which is well in line with the results of the Raman spectra and XPS. The Pd K-edge Fourier-transformed EXAFS spectra (Fig. 4c and Supplementary Table 2) of Pd@IM-S-1 exhibit an obvious peak of the Pd–Pd bond at ~2.74 Å and detected two weak peaks of the Pd–O bond (~1.99 Å) and Pd–O–Pd bond (~3.41 Å). In addition, the wavelet transform plot (Fig. 4f) of Pd@IM-S-1 show that the obvious Pd–Pd bond and a weak Pd–O bond are observed through comparing Pd foil (Fig. 4d) and PdO counterparts (Fig. 4e), respectively. The XANES, EXAFS, and wavelet transform plot results also further proved that metal Pd species (Pd$^0$) dominates the main position in the Pd@IM-S-1, but some PdO species still maintain the unreduced states. The existence of the Pd–PdO interfaces plays a critical role which might provide new active sites for the deep oxidation reaction owing to the strong interfacial interaction at the metal-oxide interfaces[37,43–45].

Based on the above conjecture, the DFT calculations are performed to insight into whether the presence of Pd−PdO interfaces is beneficial for the deep oxidation of light alkanes. As is well-known, the cleavage of the first C−H bond is the rate-determining step in the deep oxidation of light alkanes. The existence of oxygen vacancies is conducive to the activation of gaseous oxygen molecules to generate reactive oxygen species, which contribute to the cleavage of C−H bonds. Therefore, the oxygen vacancy formation energy ($E_v$) of three different models including PdO (101), Pd (111), and Pd−PdO interface has been investigated. Taking methane combustion as an example, meanwhile, we have also studied the activation energy ($E_a$) of the methane first C−H cleavage over the three different models. The detailed theoretical calculation method and related models are presented in the Supplementary Information (Supplementary Figs. 24–30). As can be seen Supplementary Figs. 26–30, the Pd−PdO interfaces show the lowest oxygen vacancy formation energy ($E_v = 1.38$ eV) and lowest activation energy ($E_a = 0.52$ eV) of the methane first C−H cleavage among these models. The results indicated that oxygen molecules are

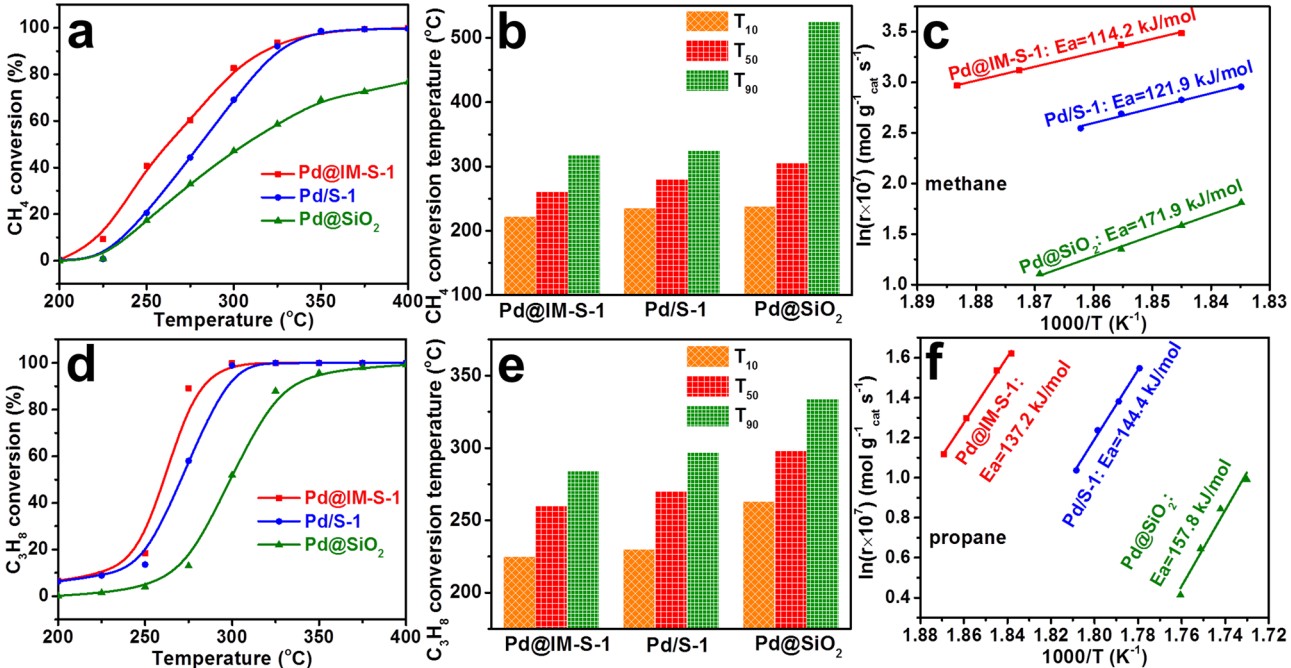

**Fig. 5 Reaction performance of Pd@IM-S-1 and related catalysts. a**, **b** Deep oxidation performances of methane, **d**, **e** deep oxidation performances of propane; **c** Arrhenius plots of methane oxidation, and **f** Arrhenius plots of propane oxidation with weight hourly space velocity (WHSV) of 180,000 mL $g_{cat.}^{-1}$ $h^{-1}$.

more easily activated on Pd−PdO interfaces and beneficial to the first C−H cleavage of light alkanes. The subsequent performance tests and in situ diffuse reflectance infrared Fourier-transform spectroscopy (in situ DRIFTS) characterization will further prove that the Pd–PdO interfaces show the highly catalytic activity for methane and propane and can provide the active lattice oxygen species ($O^-$) to oxidation of propane (detailed in situ DRIFTS analysis will be discussed soon).

**Deep oxidation performance**. The catalytic activities of the Pd@IM-S-1 and related samples for methane and propane deep oxidation are presented in Fig. 5. As displayed in Fig. 5a, d, the methane and propane oxidation activities are higher over Pd@IM-S-1 than over Pd/S-1 and Pd@SiO₂. Significantly, the catalytic activity of Pd@IM-S-1 is greatly improved compared with Pd@SiO₂. The improvement might be due to the introduction of mesopores and the higher surface areas, which expose more active centers and improve the diffusion efficiency of the reactants and products. In addition, the catalytic performance of the reduced sample (Pd@IM-S-1) and unreduced sample (PdO@IM-S-1) for methane and propane oxidation are compared. As shown in Supplementary Fig. 31, the methane and propane oxidation activities of Pd@IM-S-1 are higher than PdO@IM-S-1 indicating that the activity of metal Pd species ($Pd^0$) are higher than PdO species ($Pd^{2+}$) and the existence of Pd−PdO interfaces act a key role to provide active oxygen species for the deep oxidation of methane and propane. Moreover, reaction temperatures corresponding to methane and propane conversion at 10, 50, and 90% (noted as $T_{10}$, $T_{50}$, and $T_{90}$, respectively) are used to compare the catalytic performances of Pd@IM-S-1 and related catalysts, as displayed in Fig. 5b, e and Supplementary Table 3. It can be seen that Pd@IM-S-1 had the lowest $T_{10}$, $T_{50}$, and $T_{90}$ for methane and propane, e.g., its $T_{90}$ was as low as 318 and 284 °C for methane and propane, respectively. These results clearly reveal that methane and propane deep oxidation activities over Pd@IM-S-1 are superior to those over Pd/S-1 and Pd@SiO₂. Furthermore, CH₄ or C₃H₈ catalytic activities over Pd-based catalysts were compared with literatures

(Supplementary Table 4), which indicated that Pd@IM-S-1 is one of the best catalysts for CH₄ and C₃H₈ deep oxidation over the Pd based catalysts.

Kinetic testing was conducted to further investigate the excellent catalytic performance of Pd@IM-S-1 under high weight hourly space velocity (WHSV = 180,000 mL $g_{cat.}^{-1}$$h^{-1}$) at low conversion (<15%). A series of preliminary experiments and mathematical calculations were carried out to ensure whether the internal and external diffusion have been eliminated in intrinsic kinetic test. The Weisz−Prater criterion ($C_{WP}$) and Mears' criterion ($C_M$) were applied as the criterion, universally, if $C_{WP} < 1$ and $C_M < 0.15$, the internal and external diffusion effects could be neglected. Based on the calculation results (Supplementary Tables 5–8), it can be seen that the internal and external diffusion effects could be neglected[46,47]. As shown in Fig. 5c, f and Supplementary Table 9, it is evident that the apparent activation energies for methane and propane over Pd@IM-S-1 were the lowest among the three catalysts. Meanwhile, the reaction rates and turnover frequency values for methane and propane oxidation, calculated at 265 °C for Pd@IM-S-1, are two times higher than those over Pd/S-1 and seven times those over Pd@SiO₂. Thus, Pd@IM-S-1 displays superior deep oxidation performance compared with Pd/S-1 and Pd@SiO₂, indicating that Pd@IM-S-1 is a promising catalyst for deep oxidation of light alkanes.

Additionally, to investigate the valence change of Pd species over Pd@IM-S-1 during oxidation of light alkanes, the XRD patterns, Raman, and XPS spectra are performed for the sample of after propane oxidation reaction (noted as Pd@IM-S-1-used). As can be seen from Supplementary Fig. 32a, a new diffraction peak was detected over the Pd@IM-S-1-used sample, which could be attributed to PdO (101) lattice plane and the diffraction peak intensity is lower than unreduced sample PdO@IM-S-1. The results show that some of the reduced metal Pd species are oxidized to PdO species during the oxidation process of propane. Similarly, the Raman spectra detected a new peak over Pd@IM-S-1-used at 648 cm⁻¹ was ascribed to the typical B₁g

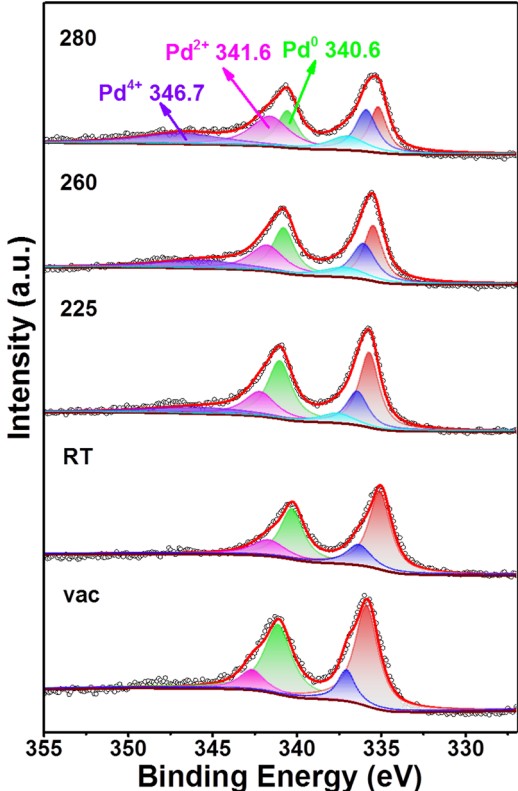

**Fig. 6 Pd speciation followed by the in situ NAP-XPS.** Pd 3d core-line spectra as a function of reaction conditions for Pd@IM-S-1 at different temperature (from room temperature to 280 °C). The total pressure in the NAP cell was fixed at 1 mbar ($C_3H_8:O_2 = 1:5$).

modes of Pd−O bond shown in Supplementary Fig. 32b[35,36]. The peak intensity of Pd−O bond over Pd@IM-S-1-used was lower than PdO@IM-S-1 consistent with XRD results. Furthermore, the XPS spectra of Pd@IM-S-1-fresh and Pd@IM-S-1-used more intuitively show the valence changes of Pd species as displayed in Supplementary Fig. 33. Compared with Pd@IM-S-1-fresh (Supplementary Fig. 33a), the peak intensity of $Pd^{2+}$ species are increasing and the peak intensity of $Pd^0$ species are weakened over Pd@IM-S-1-used (Supplementary Fig. 33b). It was further proved that some of reduced metallic Pd species are oxidized to PdO species during the oxidation process of propane.

In order to further clearly study the changes of chemical valence of Pd species during the oxidation process. The in situ near ambient pressure XPS (in situ NAP-XPS) was performed as shown in Fig. 6. It can be seen that Pd species have two states ($Pd^0$ and $Pd^{2+}$) under vacuum condition indicating some PdO species are still unreduced. The valence of Pd species remains unchanged at room temperature when the $C_3H_8$ and $O_2$ mixed gas ($C_3H_8:O_2 = 1:5$) was introduced. As the reaction temperature increasing to 225 °C, a new peak was detected around 346.7 eV in the Pd3d$_{3/2}$ spectra could be attributed to $Pd^{4+}$ species[48,49]. The peak intensity of $Pd^{2+}$ and $Pd^{4+}$ species are gradually increased and the peak intensity of $Pd^0$ species are gradually decreased as the reaction temperature continues to increase. It was proved that some of the reduced metallic Pd species are oxidized to $PdO_x$ species. The above ex situ XRD, Raman, XPS, and in situ NAP-XPS results indicate that the changes of chemical valence of Pd species are from low-valence $Pd^0$ to high-valence $Pd^{2+}$ and $Pd^{4+}$ during the oxidation process of propane. The formation and the co-existing of the Pd−PdO interfaces can efficiently provide the active oxygen species.

**Stability of Pd@IM-S-1**. Generally, the service life of a catalyst is an important indicator for evaluating its suitability for practical applications. Thermal stability, water resistance, and recycling tests were performed to investigate the durability of Pd@IM-S-1 and Pd/S-1, and propane deep oxidation was selected as the model reaction for these tests. The results are presented in Fig. 7 and the detailed measuring processes are presented in the Supplementary Information. As presented in Fig. 7a, thermal stability testing was carried out to study the anti-sintering performance of Pd@IM-S-1 and Pd/S-1. Pd@IM-S-1-800 and Pd/S-1-800 represent samples after high-temperature treatment at 800 °C for 8 h in argon. The activity of Pd@IM-S-1-800 was slightly decreased compared with the fresh sample of Pd@IM-S-1. In contrast, the activity of Pd/S-1-800 showed a significant decrease compared to the fresh sample Pd/S-1. Figure 7b presents the XRD patterns of Pd@IM-S-1-800, Pd@IM-S-1, Pd/S-1-800, and Pd/S-1. The diffraction intensities of Pd species in fresh and thermal-treated Pd@IM-S-1 are similar. However, the diffraction intensity of Pd in Pd/S-1-800 shows a significant increase over that in the fresh reduced Pd/S-1. Moreover, the Tomogram-section TEM and BET of Pd@IM-S-1-800 were further performed as shown in Supplementary Fig. 34 and Table 1. As can be seen in Supplementary Fig. 34, the Pd species still maintain relatively small particles (5.6 nm) even though the Pd species growth to larger particles (from 2.4 nm and 4.0 nm to 5.6 nm) after high temperature treatment. Most of the Pd NPs still confined within zeolite shell and homogeneously distributed over all of the zeolite. The specific surface area (Supplementary Table 1) of Pd@IM-S-1-800 was slightly decreased (from 408 m² g⁻¹ decreased to 385 m² g⁻¹), which may be due to the aggregation of Pd species leading to the blockage of some pores tunnel after high temperature treatment. The Tomogram-section TEM and BET results demonstrate that Pd@IM-S-1 have good high temperature thermal stability due to the confinement effect of the zeolite shell.

Water resistance testing over Pd@IM-S-1 and Pd/S-1 for deep oxidation of propane was used to evaluate the tolerance to water of the catalysts, as displayed in Fig. 7c. It can be seen that the activity decline trend of Pd@IM-S-1 is lower than that of Pd/S-1 at temperatures < 300 °C when 5% $H_2O$ is introduced. The effect of water on the activity of Pd@IM-S-1 and Pd/S-1 gradually decreases at temperatures > 300 °C. These tests indicate that Pd@IM-S-1 is more resistant to water than Pd/S-1, probably because of the hydrophobic and guarding effect of the zeolite shell, which prohibits the direct exposure of the active components to water vapor. As shown in Fig. 7d, the five cycling tests for $C_3H_8$ deep oxidation over Pd@IM-S-1 were performed under the same conditions. Significantly, the conversion rate at 300 °C held steady at 90% after five cycling tests (Fig. 7e), and the temperature of $T_{90}$ was slightly increased from 289 to 300 °C. Additionally, a fresh sample (Pd@IM-S-1-fresh), after the fifth cycling test (Pd@IM-S-1-5th-run), was subjected to an ordinary regeneration process (reduced at 400 °C for 2 h in 10% $H_2$−Ar, termed Pd@IM-S-1-restore), as shown in Fig. 7f. It was found that the activity over Pd@IM-S-1-restore was completely regenerated. However, the propane oxidation activity was sharp decline over Pd/S-1-5th-run after the fifth cycling test and the activity over Pd/S-1-restore cannot be regenerated (Supplementary Fig. 35). These results demonstrated that Pd@IM-S-1 has better recyclability than Pd/S-1.

**Reaction mechanism of $C_3H_8$ deep oxidation over Pd@IM-S-1.** The investigation of the reaction mechanism is significantly and meaningful, which is helpful to provide an advantageous reference basis for the design of other high-performance catalysts. Propane was selected as the model compound to investigate the deep oxidation

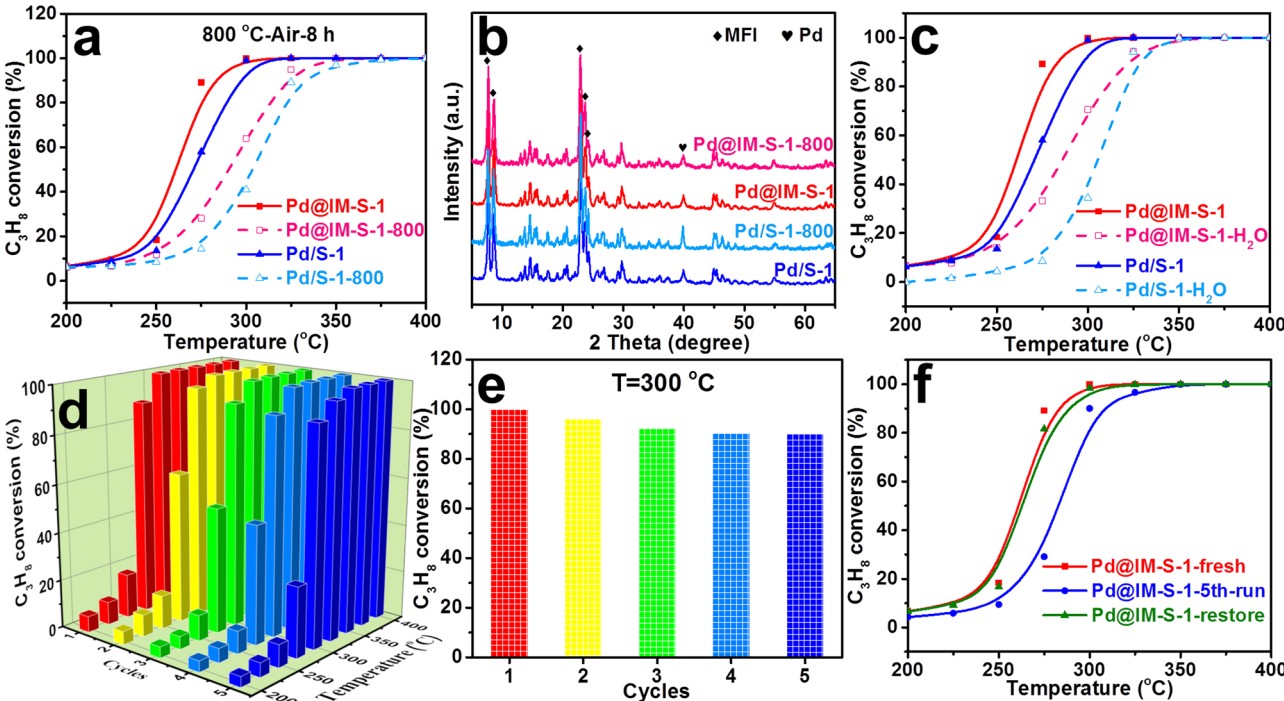

**Fig. 7 Stability test of Pd@IM-S-1 and related catalysts. a** High temperature thermal resistance, and **b** XRD patterns of Pd@IM-S-1 and Pd/S-1 before and after thermal treated at 800 °C for 8 h in argon; **c** $H_2O$ resistance of Pd@IM-S-1 and Pd/S-1 in the oxidation of propane; **d, e** recycling test of Pd@IM-S-1 in the oxidation of propane and activity at 300 °C; **f** regeneration test of Pd@IM-S-1 in the oxidation of propane.

mechanism over Pd@IM-S-1 by in situ diffuse reflectance infrared Fourier transform spectroscopy (in situ DRIFTS). We designed three types of in situ DRIFTS exploration experiments, according to literature reports[50], to obtain deeper insight into the reaction pathway of $C_3H_8$ deep oxidation over Pd@IM-S-1, that is $C_3H_8$ adsorption and total oxidation experiments, $C_3H_8$-TPD experiments in $N_2 + O_2$ feed gases, and $C_3H_8$ temperature-programmed oxidation experiments. The detailed measuring processes are described in the Supplementary Information. According the in situ DRIFTS results (Supplementary Figs. 36–38 and Table 10, the detailed analysis was presented in the Supplementary Information), it can be concluded that propane deep oxidation over Pd@IM-S-1 might follow the Mars-van-Krevelen (MvK) mechanism involving the active oxygen species from Pd–PdO interfaces, which participate in the redox catalytic cycle. The acetate and bicarbonate species can be considered as the active intermediates, and the aliphatic ester, acetone, and formate species can be considered as the inert intermediates. For the whole oxidation process, the active intermediate species and the inert intermediate species from the initial degradation of propane occupied the active sites. Their presence limited the access of the gaseous oxygen species to the active sites because there is a competitive adsorption relationship between these species. Therefore, the gaseous oxygen species can fill the oxygen vacancies or defect sites preferentially, further providing reactive oxygen species for the propane oxidation reaction. The possible reaction pathway of $C_3H_8$ deep oxidation of Pd@IM-S-1 is illustrated in Supplementary Fig. 39. First, gaseous $C_3H_8$ is adsorbed on the Pd active sites and oxidized to an acetone intermediate via active oxygen of Pd–PdO interfaces. Then the acetone intermediate is degraded to an acetate intermediate by the active oxygen species. Subsequently, the acetate intermediate is decomposed to bicarbonate and formate intermediates by the active oxygen. Finally, the bicarbonate and formate intermediates are transformed to $H_2O$ and $CO_2$ by the active oxygen, while the gaseous $O_2$ fills the vacancies of the Pd–PdO interfaces.

In summary, a facile two-step in situ mesoporogen-free method was developed to prepare a single crystal intra-mesopore zeolite confined Pd NPs (Pd@IM-S-1). The as-synthesized Pd@IM-S-1 catalyst exhibited excellent methane and propane deep oxidation performances compared with Pd/S-1 prepared by the conventional impregnation method. The high thermal stability, hydrothermal stability, and recyclability of Pd@IM-S-1 can be attributed to the guarding effect of the S-1 zeolite shell. The uniform mesoporous structure and the nanosized crystals of the zeolite shell promoted the mass-transfer efficiency and active site accessibility. The Pd–PdO interfaces as a new active site can provide active oxygen species to the first C−H cleavage of deep oxidation light alkanes, which are beneficial to improve the catalytic performance. Importantly, the designed hydrophobic zeolite shell could facilitate the adsorption of the hydrophobic light alkanes and desorption of the water produced in deep oxidation, which enhances its activity and water resistance. The facile two-step in situ mesoporogen-free strategy developed in this work to synthesize the single-crystalline intra-mesopore zeolite and simultaneously confine the Pd NPs catalyst (Pd@IM-S-1) follows the dissolution–recrystallization mechanism and has the universality to confine other metal NPs (Pt, Rh, Ru, et.al.), and therefore provides a guideline to design other high-performance intra-mesopore zeolite confined nano-catalysts for high-temperature environmental or thermal catalysis.

## Methods

**Catalyst synthesis**. Pd@IM-S-1 was synthesized by a facile two-step in situ dry-gel conversion method without adding any mesopore-creating agent (mesoporogen-free) using the porous silica confined Pd as the precursor. For comparison, Pd NPs supported on the outer surface of S-1 were prepared by the conventional impregnation method. To reveal the formation mechanism of Pd@IM-S-1, the samples crystallized at each step were collected. Furthermore, to verify the universality of the facile in situ mesoporogen-free method, mesoporous zeolite confined other metal active sites were also prepared. The detailed preparation procedures are discussed in the Supplementary Information.

**Characterization methods**. Various testing methods—including XRD, Raman spectra, ICP-OES, XPS, scanning electron microscopy (SEM), TEM, HRTEM, energy-dispersive X-ray spectroscopy (EDS) elemental mapping, tomogram-section TEM, FIB, aberration-corrected TEM (AC-TEM), AC-EDS elemental

mapping, XANES, EXAFS, in situ near ambient pressure XPS (in situ NAP-XPS), $N_2$ adsorption/desorption analysis and the in situ diffuse reflectance infrared Fourier-transform spectroscopy (in situ DRIFTS)—were adopted to measure the physical and chemical performance and plausible reaction mechanism over Pd@IM-S-1 and related samples. The detailed measuring processes are described in detail in the Supplementary Information.

**Activity and kinetic tests**. Methane ($CH_4$) and propane ($C_3H_8$) deep oxidation: Methane and propane, as typical light alkanes, were selected as the model compounds for testing the catalytic performance of Pd@IM-S-1. The detailed activity and kinetic tests are also presented in the Supplementary Information.

**Density functional theory calculations**. The formation energy of oxygen vacancy ($E_v$), the activation energy ($E_a$) of the methane first C−H cleavage with three different models including PdO (101), Pd (111) and Pd−PdO interface are studied by DFT calculations. The details about the theoretical calculation method can be found in the Supplementary Information.

## Data availability

The additional data are provided in the Supplementary Information. All the data that support the findings of this study are available from the corresponding author upon reasonable request. Source data are provided with this paper.

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

## Acknowledgements

Support for this work from the National Natural Science Foundation of China (21976078, 21922606, 21876139, and 21773016), the National Key R&D Program of China (2016YFC0205900), and the Natural Science Foundation of Jiangxi Province (20202ACB213001) is greatly acknowledged by H.P. S.D. was supported by the Division of Chemical Sciences, Geosciences, and Biosciences, Office of Basic Energy Sciences, US Department of Energy. Dr. An also want to express his thanks for financial support from Guangdong Provincial Key R&D Program (2019B110206002).

## Author contributions

H.P. and S.D. conceived the research idea and designed the experiments. T.D. and H.P. performed all the experiments and analyzed all the data. S.Y., W.L., Y.P., and X.C. took part in the synthesis of samples and characterizations. J.T. performed the DFT calculation. H.C., Z.Y., C.H., P.W., D.W., T.A., and Y.W. discussed the results and commented on the manuscript. H.P., T.D., H.C, Z.Y., and S.D. co-wrote and revised the paper.

## Competing interests

The authors declare no competing interests.
