## [Peer Review File · Nature Communications]

Title: Intra-crystalline mesoporous zeolite encapsulation-derived thermally robust metal nanocatalyst in deep oxidation of light alkanesREVIEWER COMMENTS

Reviewer #1 (Remarks to the Author):

Review of "In situ intra-crystalline mesoporous zeolite encapsulation-derived thermally roust metal nanocatalyst in deep oxidation" by H. Peng et al.

The manuscript of H. Peng et al. reports on the synthesis of Pd-based catalysts for the oxidation of light alkane (methane, propane) to CO₂ and H₂O, based on encapsulated Pd nanoparticles in zeolite crystals (silicalite-1) with tuned mesoporosity for better mass transport and Pd thermal stabilization. While the work reported is scientifically sound and the proposed synthesis strategy is interesting for catalysts design, the work lacks of new fundamental insights and does not meet the novelty required for publication in Nature Communications.

Pd thermal stabilization by encapsulation is a well-established strategy in heterogeneous catalysis (e.g. Science 2012, 337, 713; Nat. Commun 2019, 10, 1611), leading to high activity at low temperature in methane oxidation, comparable to those reported herein. Control of zeolite porosity aimed at metal encapsulation and stabilization is also a well-established field (e.g. Front Chem. 2018, 6, 550). The X-ray absorption results and in-situ infrared spectroscopy reported herein also do not shed new light into the chemistry of Pd under deep oxidation of alkanes or into the reaction mechanism (e.g. ACS Catal. 2015, 5, 4, 2481–2489; Nat. Commun 2018, 9, 2545; Renew. Sust. Energ. Rev., 2020, 119, 109589).

I therefore think the manuscript is not suitable for publication in Nature Communication and might be suitable for publication in another journal, such as ACS Catalysis. Some comments follow:

1. I would encourage the authors to proof-read the manuscript before resubmission: there is a typo even in the title ("roust" instead of "robust"). The title is also very obscure and should be rephrased (e.g. specifying "deep oxidation of alkanes", removing "in situ" from the title, etc.)
2. I suggest the authors to make the writings in all the figures much bigger, since they are very hard to read when printed on paper.
3. Figure 1 is not very clear, and is more fit for a table of content than a synthesis scheme.
4. Figure 3 is also not very dense with information. I would suggest to couple it with some evidence to support the proposed mechanism of mesopores formation.

Reviewer #2 (Remarks to the Author):

In recent years, the zeolite-confined metal nanoparticles (NPs) have attracted much attention owing to their superior sintering resistance and broad applications for high-temperature thermal and environmental catalytic reactions. However, pore sizes of the conventional zeolites are usually below 2 nm, and the reactants are easily blocked to access the active metal sites. Therefore, the design and synthesis of confined catalysts with meso- or hierarchical porous structures are urgent. In this

manuscript, Peng et. al. developed a facile in situ mesopore-free strategy to design and synthesize palladium NPs enveloped within a single-crystalline zeolite (silicalite-1, S-1) with intra-mesopores (Pd@IM-S-1), and importantly, Pd@IM-S-1 exhibited remarkable activity for light alkanes deep oxidation, sintering and water resistance (even calcined at 800 °C for 8 h), thermal stability, and recyclability. The authors make a great contribution to investigate the formation mechanism, method universality and catalytic performance of the novel catalysts. Thereby I recommend acceptance of this work for publication after minor revisions have been made according the following comments:

1. The scale bar in Figure 2 should be made more clearly, e.g., using the thicker white line.
2. FIB technique is one of the audio-visual methods to confirm the confined structure. The authors need to provide the measuring procedures in detail.
3. As a hierarchical porous material, its external specific surface area needs to be provided in Table S1.
4. Generally, the kinetic test should be carried out before excluding the mass transportation limits. Thus, I suggest the authors provide a detailed kinetic calculation to confirm that their kinetic results were obtained after excluding the mass transportation limits.
5. It is well known that the Pd or PdOx species have activities for the addressed reactions. While the results provided in this manuscript suggest that the Pd-PdO interface are the active sites. The authors need to provide some important literature to support their results.
6. Some of the important Figures or Scheme in the Supporting Information need to be moved to the main text.

Response to Reviewers' Comments

Ms. Ref. No.: NCOMMS-21-02997

Title: In situ intra-crystalline mesoporous zeolite encapsulation-derived thermally robust metal nanocatalyst in deep oxidation

Reviewers' comments:

Reviewer #1 (Remarks to the Author):

Review of "In situ intra-crystalline mesoporous zeolite encapsulation-derived thermally robust metal nanocatalyst in deep oxidation" by H. Peng et al.

The manuscript of H. Peng et al. reports on the synthesis of Pd-based catalysts for the oxidation of light alkane (methane, propane) to CO₂ and H₂O, based on encapsulated Pd nanoparticles in zeolite crystals (silicalite-1) with tuned mesoporosity for better mass transport and Pd thermal stabilization. While the work reported is scientifically sound and the proposed synthesis strategy is interesting for catalysts design, the work lacks of new fundamental insights and does not meet the novelty required for publication in Nature Communications.

Reply:

Thanks for your careful and hard work to review our manuscript. As we all know, noble metal-based materials are considered one of the most efficient catalysts for deep oxidation of VOCs because of their high activity. However, at the present time, traditional supported noble metal-based catalysts still have some challenges in practical applications due to their bad thermal stability at high thermal reaction temperatures. Recently, zeolite-confined metal nanoparticles (NPs) have attracted much attention owing to their superior sintering resistance and broad applications for high-temperature thermal and environmental catalytic reactions.

Although there are already lots of reported literature works about zeolite-confined metal nanoparticles (NPs) catalysts and used in various catalytic reactions, such as, **Prof. Avelino Corma** from Instituto de Tecnología Química reported "*Generation of*

subnanometric platinum with high stability during transformation of a 2D zeolite into 3D” published in **Nature Materials** (**Nat. Mater.** 2017, 16, 132-138), and “*Regioselective generation and reactivity control of subnanometric platinum clusters in zeolites for high-temperature catalysis*” published in **Nature Materials** (**Nat. Mater.** 2019, 18, 866-873), and “*Structural modulation and direct measurement of subnanometric bimetallic PtSn clusters confined in zeolites*” published in **Nature Catalysis** (**Nat. Catal.** 2020, 3, 628-638), etc.; **Prof. Xiao** from Zhejiang University reported “*Sinter-resistant metal nanoparticle catalysts achieved by immobilization within zeolite crystals via seed-directed growth*” published in **Nature Catalysis** (**Nat. Catal.** 2018, 1, 540-546), and “*Hydrophobic zeolite modification for in situ peroxide formation in methane oxidation to methanol*” published in **Science** (**Science**, 2020, 367, 193-197); **Prof. Yu** from Jilin University reports “*In situ confinement of ultrasmall Pd clusters within nanosized silicalite-1 zeolite for highly efficient catalysis of hydrogen generation*” published in **JACS** (**J. Am. Chem. Soc.** 2016, 138, 7484-7487), and “*Subnanometric hybrid Pd-M(OH)₂, M=Ni,Co, clusters in zeolites as highly efficient nanocatalysts for hydrogen generation*” published in **Chem** (**Chem** 2017, 3, 477-493), etc. These zeolite-confined metal nanoparticles (NPs) catalysts reported by these researchers have shown excellent catalytic performance and their nice work greatly contributed to the design of high-performance catalysts.

However, there are still some existing problems for these microporous zeolites confined catalysts because the pore size of the conventional zeolites is usually below 2 nm and limits the molecule diffusion and the metal active sites accessibility, making it unfavorable for the diffusion of the reactants and products and exposing of the metal active sites. **Therefore, promoting the mass-transfer efficiency of reactants and products and accessibility of the active sites over the conventional zeolite confined catalysts is an urgent problem for various catalytic reactions, particularly, the deep oxidation of VOCs. At present, a potential good strategy to resolve these problems is to synthesize a mesoporous-zeolite shell to promote the mass-transfer efficiency and accessibility of the active sites.** There are many ways to obtain mesoporous zeolites, which can be roughly divided into the following four categories:

(1) using removable and inert solids as hard templates, (2) using amphiphilic surfactants as soft templates, (3) using post-synthetic treatment for demetallization (e.g., Si, Al), and (4) assembling zeolite nanosheets or nanocrystals. **Although these methods can produce mesoporous zeolites, to the best of our knowledge, it is still a challenge to fabricate single-crystal zeolites with intra-crystalline mesopores (intra-mesopores) and confine the active metal NPs simultaneously via a simple one-pot in situ strategy.** Because it is still difficult to obtain mesoporous zeolites, and some disadvantages are not overcome such as high costs, health hazards, limitations on the types of synthetic zeolites, and decreased crystallinity of the zeolites.

In this work, we developed a facile in situ mesopore-free strategy to design and synthesize palladium (Pd) NPs enveloped within a single-crystalline zeolite (silicalite-1, S-1) with abundant intra-mesopores (termed Pd@IM-S-1). The synthesis steps are as follows: (1) using a reversed-phase microemulsion method obtained the Pd@SiO₂ precursor, (2) the Pd@IM-S-1 sample was prepared by in situ conversion of the amorphous SiO₂ shell of Pd@SiO₂ to silicalite-1 shell through a continuous heating in situ dry-gel conversion method without the involvement of additional mesopore-creating agents. **Our work was different from the above-mentioned literature works which are only containing the single microporous channels, and our work contains micropores and the supernumerary intra-mesopores simultaneously, and the synthetic method was compared with the previous works as shown in Scheme S1. The introduction of mesopores is more conducive to the exposure of active sites, and importantly, can promote the mass-transfer efficiency of the reactants and products and the accessibility of the metal active sites.**

Scheme S1. Comparison the previous synthetic pathways of zeolite-confined metal nanoparticles (NPs) catalysts.

Additionally, we are further investigating the reactivity of Pd@IM-S-1 for toluene oxidation as shown in **Figure R1**. Interestingly, the Pd@IM-S-1 sample is also showed better toluene oxidation performance than Pd/S-1 sample. **These results indicate that**

the mesoporous zeolite confined NPs catalysts are also suitable for other reactions, which can be attributed to the introduction of mesopores can improve the mass-transfer efficiency of reactants and products and the accessibility of the metal active sites. Meanwhile, we have further investigated the universality of the facile in situ mesoporogen-free method, and successfully prepared the Pt@IM-S-1, Rh@IM-S-1, and Ru@IM-S-1 catalysts. Thereby, this simple in situ mesoporogen-free strategy exhibits versatility and potential applicability, and also provides a guideline to design other high-performance intra-mesopore zeolite confined nanocatalysts for high-temperature environmental or thermal catalysis. Thus, we believe that the novelty of this manuscript can bring about an important impact in environmental catalysis and air pollution control areas.

Figure R1. Deep oxidation performances of toluene over Pd@IM-S-1 and Pd/S-1. Conditions: 1000 ppm C₇H₈ is produced by bubbling with Air and the WHSV was 60000 mL g_{cat.}⁻¹ h⁻¹.

According to the reviewer's suggestions, the aforementioned discussion has been added to the main text to explain the novelty of this manuscript.

Pd thermal stabilization by encapsulation is a well-established strategy in heterogeneous catalysis (e.g. *Science* 2012, 337, 713; *Nat. Commun* 2019, 10, 1611), leading to high activity at low temperature in methane oxidation, comparable to those reported herein.

Reply:

Thanks for your careful and hard work to review our manuscript and kindly giving constructive suggestions to us, which are very important for us to improve the quality of the previously submitted manuscript. As you know, Pd thermal stabilization by encapsulation is a well-established strategy in heterogeneous catalysis and leading to high activity at low temperature in methane oxidation. According to your suggestion, we have compared these two literature citations with our work, the Pd@CeO₂/Al₂O₃ for methane combustion (*Science* 2012, 337, 713) and the Pd/NA-Al₂O₃ for methane and propane oxidation (*Nat. Commun* 2019, 10, 1611). Actually, the difference between Pd/NA-Al₂O₃ (*Nat. Commun* 2019, 10, 1611) and Pd@IM-S-1 (our work) has been compared in the supporting information and presented in **Table S4**. In addition, we also added Pd@CeO₂/Al₂O₃ (*Science* 2012, 337, 713) to the **Table S4** for comparison and marked with yellow highlighting. These results show that the T₉₀ (reaction temperatures corresponding to methane conversion at 90%) of Pd@CeO₂/Al₂O₃ was > 350 °C, and our work (Pd@IM-S-1) was about 318 °C. Similarly, the T₉₀ of methane and propane over Pd/NA-Al₂O₃ was > 325 and ~300 °C, respectively, Pd@IM-S-1 prepared in our work was about 318 and 289 °C. Based on the above results, the catalytic performance of Pd@IM-S-1 in our work was better than that of Pd@CeO₂/Al₂O₃ and Pd/NA-Al₂O₃, and our work shows a unique advantage.

According to the reviewer’s suggestions, the aforementioned mentioned two works have been added to **Table S4** for comparison.

Table S4. Comparison with literatures about the performances of Pd based catalysts for CH₄ or C₃H₈ oxidation.

Catalysts	Pd wt. %	Reaction conditions			T ₉₀ (°C)	References
		CH ₄	C ₃ H ₈	WHSV(mL)		

		(%)	(%)	$\text{g}^{-1} \text{h}^{-1}$		
Pd@IM-S-1	1.83	1	-	36000	318	This work
		-	0.2		289	
Pd/Al-Ce(550)	1	0.1	-	50400 h^{-1}	> 400	Appl. Catal. B Environ. 264 (2020) 118475
PdSiCe	0.92	0.5	-	180000 h^{-1}	> 600	Appl. Catal. A Gen. 574 (2019) 79-86
Pd/NA-Al₂O₃	5	1	-	15000	> 325	Nat. Commun. 10 (2019) 1611
		-	0.2	30000	~ 300	
Pd/CeO ₂	1	-	0.3	240000	401	Appl. Catal. B Environ. 226 (2018) 585-595
Pd/CZ/A	2	-	0.012	40000 h^{-1}	> 300	J. Ind. Eng. Chem. 58 (2018) 246-257
Pd/CeO ₂	0.77	1	-	15000	332	Catal. Sci. Technol. 8 (2018) 2567
Pd _{0.78} Mn _{0.22}	0.5	0.5	-	20000	400	J. Am. Chem. Soc. 139 (2017) 11989-11997
Pd/Al ₂ O ₃	1.6	-	0.06	40000 h^{-1}	315	Rare Metal Mat. Eng. 46 (2017) 1231-1236
Co _{3.5} Pd/3DOM CeO ₂	0.4	2.5	-	40000	520	J. Catal. 342 (2016) 17-26
PdO/CeO ₂ @HZSM-5	0.93	0.5	-	30000	> 535	Nanoscale 8 (2016) 9621-9628
Pd/Al ₂ O ₃	3	-	0.175	300000	~ 375	Catal. Today 201 (2013) 19-24
Pd/ZSM-5	1.5	-	0.2	30000	~ 334	ACS Catal. 3 (2013) 1154-1164
Pd _y /W _y /TiO ₂	2	-	0.5	45000 h^{-1}	345	J. Catal. 285 (2012) 103-114
Pd@CeO₂/Al₂O₃	1	0.5	-	200000	> 350	Science 337 (2012), 713

Pd/V/Al ₂ O ₃	0.5	-	0.5	45000 h ⁻¹	385	Catal. Sci. Technol. 1 (2011) 1367-1375
-----	---	-----	-----------------------	-----	--

Control of zeolite porosity aimed at metal encapsulation and stabilization is also a well-established field (e.g. *Front Chem.* 2018, 6, 550).

Reply:

Thanks for your careful and hard work to review our manuscript. As you mentioned, control of zeolite porosity aimed at metal encapsulation and stabilization is also a well-established field. We agree with the reviewer's point, because of the hard work of these scientists in the area of zeolite-confined metal nanoparticles (NPs) catalysts, this method has been well established. As mentioned above, lots of excellent studies have been reported in the design of high-performance zeolite confined catalysts.

According to the existing reports, this strategy is mostly applied to microporous zeolite-confined metal nanoparticles (NPs) catalysts because it is still difficult to obtain mesoporous zeolites. We have carefully read this review (*Front Chem.* 2018, 6, 550) you mentioned and obtained the similar views as we mentioned above. There are many strategies (demetallization method, soft- and hard- template methods, etc.) that have been developed to obtain mesoporous zeolites. **In literature (*Front Chem.* 2018, 6, 550), it was reported that some disadvantages were not overcome such as high costs, health hazards, limitations on the types of synthetic zeolites, and decreased crystallinity of the zeolites. Therefore, though these methods can generate mesoporous zeolites, to the best of our knowledge, it is still a challenge to fabricate single-crystal zeolites with intra-crystalline mesopores (intra-mesopores) and confine the active metal NPs simultaneously via a simple one-pot in situ strategy.**

In our work, we developed a novel facile *in situ* mesopore-free strategy to design and synthesize palladium (Pd) NPs enveloped within a single-crystalline zeolite (silicalite-1, S-1) with abundant intra-mesopores (termed Pd@IM-S-1). The synthesis steps are as follows: (1) using a reverse-phase microemulsion method to obtain Pd@SiO₂ as precursor, (2) the Pd@IM-S-1 sample was obtained by the *in situ*

conversion of the amorphous SiO₂ shell of Pd@SiO₂ to silicalite-1 shell through a continuous heating in situ dry-gel conversion method. Our work provided a novel simple strategy without involvement of additional mesoporous creating agents or other organic ligands and is very different from the methods mentioned in the existing reports to obtain mesoporous zeolites, especially for confining the metal NPs simultaneously.

The X-ray absorption results and in-situ infrared spectroscopy reported herein also do not shed new light into the chemistry of Pd under deep oxidation of alkanes or into the reaction mechanism (e.g. ACS Catal. 2015, 5, 4, 2481–2489; Nat. Commun 2018, 9, 2545; Renew. Sust. Energ. Rev., 2020, 119, 109589).

I therefore think the manuscript is not suitable for publication in Nature Communication and might be suitable for publication in another journal, such as ACS Catalysis. Some comments follow:

Reply:

Thanks for your careful and hard work to review our manuscript. According to the current reports, fundamental or mechanism studies have revealed that the metal–oxide interface played a significant role in enhancing catalytic activity owing to strong interfacial interaction (**J. Mater. Chem. A, 2019, 7, 12627**). The interface of Pd-PdO has been reported as one of the most catalytically active species in hydrocarbons oxidation (**ACS Catal. 2017, 7, 4372-4380; Appl. Catal. B: Environ. 2018, 236, 436-444; Angew. Chem. Int. Ed. 2020, 59, 18522-18526**). Therefore, the original intention to perform the X-ray absorption near-edge structure (XANES) and the extended X-ray absorption fine structure (EXAFS) characterization test is to further prove the existence of the Pd-PdO interface. The results also proved the existence of this Pd-PdO interface, which is well in line with the results of the Raman and XPS spectra, as shown in **Figure 4** and **Figure S19**. Additionally, the *in situ* diffuse reflectance infrared Fourier-transform spectroscopy (in situ DRIFTS) characterization technique proved that the Pd–PdO interface could provide the active lattice oxygen species (O[•]) to oxidize the propane. Thus, in our opinion, the Pd–PdO interface can be seen as a new reaction site

for deep oxidation reactions. This view is similar with the literature as mentioned in the above.

Figure 4. XPS of Pd3d for Pd@IM-S-1, Pd/S-1 and Pd@SiO₂ (A); XANES spectra at the Pd K-edge (B); Fourier transform of k³-weighted EXAFS spectra at the Pd K-edge (C); the wavelet transforms from experimental data for Pd foil, PdO, and Pd@IM-S-1 (D-F).

Figure S19. Raman spectra of unreduced samples PdO@IM-S-1, PdO/S-1, PdO@SiO₂ (A) and reduced samples Pd@IM-S-1, Pd/S-1, Pd@SiO₂ (B).

Furthermore, we have carefully read the literature (ACS Catal. 2015, 5, 4, 2481–2489; Nat. Commun 2018, 9, 2545; Renew. Sust. Energ. Rev., 2020, 119, 109589) you

mentioned. The works of “**ACS Catal. 2015, 5, 4, 2481–2489**” and “**Nat. Commun 2018, 9, 2545**” are both investigating the changes of chemical valence of Pd species through **in-situ X-ray absorption spectroscopy** method. In our work, we adopted the **ex-situ X-ray absorption spectroscopy** technology due to the limitations of current experimental conditions. Thereby, it is difficult to obtain the changes of the chemical valence of Pd species during the oxidation of alkanes process. In addition, the paper “**Renew. Sust. Energ. Rev., 2020, 119, 109589**” is a review about methane combustion, which only briefly reviewed the mechanism and kinetics of the methane combustion but did not provide deeply insights into the valence state changes of Pd species during the reaction process. We apologize that the in-situ X-ray absorption spectroscopy could not be performed at the present time. **Alternatively, some other traditional ex-situ characterization methods (such as XRD, Raman, and XPS) were adopted to verify the valence change of Pd species.**

Therefore, the valence change of the Pd species over Pd@IM-S-1 after the light alkanes (propane) deep oxidation reaction (noted as Pd@IM-S-1-used) were further characterized through the XRD patterns, Raman and XPS spectra. As can be seen from **Figure S21A**, a new diffraction peak was detected over the Pd@IM-S-1-used sample, which could be attributed to PdO (101) lattice plane and the diffraction peak intensity is lower than the unreduced sample PdO@IM-S-1. The results show that some of reduced metallic Pd species are oxidized to PdO species during the oxidation process of propane. Similarly, the Raman spectra detected a new peak over Pd@IM-S-1-used at 648 cm^{-1} was ascribed to the typical B_{1g} modes of Pd-O bond shown in **Figure S21B**. The peak intensity of Pd-O bond over Pd@IM-S-1-used was lower than PdO@IM-S-1 consistent with the XRD results. Furthermore, the XPS spectra of Pd@IM-S-1-fresh and Pd@IM-S-1-used showing the valence changes of Pd species are displayed in **Figure S22**. Compared with Pd@IM-S-1-fresh (**Figure S22A**), the peak intensity of Pd^{2+} species are increased and the peak intensity of Pd^0 species are weakened over Pd@IM-S-1-used (**Figure S22B**). It was further proved that some of the reduced metallic Pd species are oxidized to PdO species during the oxidation process of propane. The above XRD, Raman and XPS results indicate that the changes of chemical valence

of Pd species are from Pd⁰ to Pd²⁺ during the oxidation process of propane.

Figure S21. XRD patterns (A) and Raman spectra (B) of unreduced sample PdO@IM-S-1, unreacted sample Pd@IM-S-1-fresh, and reacted sample Pd@IM-S-1-used.

Figure S22. XPS of Pd3d for Pd@IM-S-1-fresh (A) and Pd@IM-S-1-used (B).

MAJOR point:

Comment 1:

I would encourage the authors to proof-read the manuscript before resubmission: there is a typo even in the title (“roust” instead of “robust”). The title is also very obscure and should be rephrased (e.g. specifying “deep oxidation of alkanes”, removing “in situ” from the title, etc.).

Reply:

Thanks for your careful and hard work to review our manuscript and kindly giving

constructive suggestions to us, which were very important for us to improve the quality of the previously submitted manuscript. According to your suggestion, we have corrected the typos and grammatical errors, and the title of the manuscript has also been rephrased. The corrections in the whole manuscript are marked with yellow highlighting. Thanks again for your valuable comments!

Such as:

(1) The title of “**In situ intra-crystalline mesoporous zeolite encapsulation-derived thermally roust metal nanocatalyst in deep oxidation**” was revised to “**Intra-crystalline mesoporous zeolite encapsulation-derived thermally robust metal nanocatalyst in deep oxidation of light alkanes**”.

(2) “Pd@IM-S-1 exhibited remarkable light alkanes deep oxidation activity, sintering and water resistance (even calcined at 800 °C for 8 h), thermal stability, and recyclability.” was revised to “Pd@IM-S-1 exhibited remarkable light alkanes deep oxidation activity, anti-sintering (even calcined at 800 °C for 8 h) and water resistance, and recyclability.”

(3) “Physicochemical properties of Pd@IM-S-1 catalysts” was revised to “Synthesis and physicochemical properties of Pd@IM-S-1”

(4) “The elemental EDS-mapping images evidently demonstrate that the Pd element are homogeneously distributed over all the zeolite.” was revised to “The elemental EDS-mapping images evidently demonstrate that the Pd element is homogeneously distributed over all of the zeolite.”

(5) “....., the detailed experimental process are presented in the Supplementary Information.” was revised to “....., the detailed experimental process is presented in the Supplementary Information.”

(6) “.....propane oxidation, calculated at 265 °C over Pd@IM-S-1,.....” was revised to “.....propane oxidation, calculated at 265 °C for Pd@IM-S-1,.....”

(7) “It was demonstrated that C₃H₈ oxidation occurred before the O₂ feed gas introduced, demonstrating that.....” was revised to “It was demonstrated that C₃H₈ oxidation occurred before the O₂ feed gas was introduced, demonstrating that.....”

(8) “.....propane deep oxidation over Pd@IM-S-1 might follow the Mars-van

Krevelen (MvK) mechanism.....” was revised to “.....propane deep oxidation over Pd@IM-S-1 might follow the Mars-van-Krevelen (MvK) mechanism.....”

Comment 2:

I suggest the authors to make the writings in all the figures much bigger, since they are very hard to read when printed on paper.

Reply:

Thanks for your constructive suggestions. According to your suggestion, the words and numbers of in all the figures are both changed to much bigger in the revised manuscript.

Comment 3:

Figure 1 is not very clear, and is more fit for a table of content than a synthesis scheme.

Reply:

Thanks for the suggestions. According to your suggestion, we provided a higher resolution version of **Figure 1**. And the **Figure 1** is placed in the results section as a table of contents, and we are also changed the title of **Figure 1**.

Figure 1. Schematic illustration for synthesis of Pd@IM-S-1 and application for deep oxidation light alkanes.

Comment 4:

Figure 3 is also not very dense with information. I would suggest to couple it with some evidence to support the proposed mechanism of mesopores formation.

Reply:

Thanks for your constructive suggestions. We will do our best to provide clear pictures to meet the high standard of Nature Communications. Actually, to understand the formation mechanism of Pd@IM-S-1, the crystallinity and morphological evolution at different crystallization time has been monitored by XRD and TEM which are presented in the Supplementary Information (**Figure S11-S18**). And the detailed process is also illustrated in our original text. According to your constructive suggestions, in order to make this formation mechanism clearer, we have recombined the **Figure S11-S18** and the **original-Figure 3** into a **new Figure 3**. Furthermore, we have re-explained its mesopores formation mechanism of Pd@IM-S-1 as shown below.

Figure 3. TEM image of Pd@SiO₂ (A), TEM images of Pd@IM-S-1 crystallized 1 day

(B) and 2 days (C) at 80 °C; TEM images of Pd@IM-S-1 crystallized for 1 h (D), 2 h (E), 4 h (F), 12 h (G), and 24 h (H) at 120 °C; XRD patterns of Pd@IM-S-1 crystallized between 80 to 120 °C at different crystallization time (I); proposed formation process (J) of the intra-mesopore over Pd@IM-S-1.

As shown in **Figure 3I**, the Pd@IM-S-1-80-1d still maintain the amorphous SiO₂ after being aged at 80 °C for one day. The TEM image of Pd@IM-S-1-80-1d demonstrated that partial dissolution of the SiO₂ shell occurred (**Figure 3B**). The Pd@IM-S-1-80-2d sample detected new diffraction peaks ascribed to MFI crystalline structure when crystallization time increased to two days at 80 °C (**Figure 3I**). The corresponding TEM image (**Figure 3C**) show further dissolution of the shell SiO₂, and spherical particles are beginning to connect to each other. The results of **Figure 3I** and **Figure 3C** indicated that the interfaces of amorphous silica-zeolite are formed. Subsequently, the crystallinity of Pd@IM-S-1 gradually increased from 1 h to 2 h at 120 °C, and the crystallinity of Pd@IM-S-1 remains unchanged after the crystallization time further increased to 4 h at 120 °C (**Figure 3I**). It is observed in **Figure 3D** that the multiple small Pd@SiO₂ particles are connected more tightly, and are beginning to reorganize into large Pd@IM-S-1 particles. The morphology is further evolved, and some mesopores are observed over Pd@IM-S-1-120-2h as shown in **Figure 3E**. From **Figure 3F-3H**, it could be observed that the quantities of mesopores are increased evidently, while the morphology remains unchanged.

Based on the results mentioned above, it can be concluded that the growth of Pd@IM-S-1 is a fast dissolution and recrystallization process, and the formation process of Pd@IM-S-1 is shown in **Figure 3J**. The microporous structure directing agent (tetrapropylammonium hydroxide, TPAOH) acting as a bifunctional reagent hydrolyzed into OH⁻ and TPA⁺, which etched the amorphous silica from Pd@SiO₂ and directed the formation of the zeolite framework. First, the silica shell of Pd@SiO₂ partially dissolved, and small particles are aggregated into bulk particles. The interstitial pores between particles appeared, due to the spherical materials close contact with each other is impossible. Subsequently, the amorphous silica-zeolite interfaces are formed

with the passage of crystallization time and the crystallinity of nanocrystals are further improved. In this process, the Si species and Pd species migrated and diffused among the amorphous silica-zeolite interfaces. However, the diffusion rates of the Si species and Pd species near the interfaces were rapid, whereas the Si and Pd species far away from the interfaces were slow to diffuse, due to the difference in diffusion rates and the presence of interstitial pores. Finally, generating mesopores and confining the Pd species in the mesopores occurred simultaneously. A similar growth mechanism was also found by other groups (**Chem. Eur. A**, 2018, 25, 738-742; **Chem. Commun.** 2015, 51, 12563-12566)

Reviewer #2 (Remarks to the Author):

In recent years, the zeolite-confined metal nanoparticles (NPs) have attracted much attention owing to their superior sintering resistance and broad applications for high-temperature thermal and environmental catalytic reactions. However, pore sizes of the conventional zeolites are usually below 2 nm, and the reactants are easily blocked to access the active metal sites. Therefore, the design and synthesis of confined catalysts with meso- or hierarchical porous structures are urgent. In this manuscript, Peng et. al. developed a facile in situ mesoporogen-free strategy to design and synthesize palladium NPs enveloped within a single-crystalline zeolite (silicalite-1, S-1) with intramesopores (Pd@IM-S-1), and importantly, Pd@IM-S-1 exhibited remarkable activity for light alkanes deep oxidation, sintering and water resistance (even calcined at 800 °C for 8 h), thermal stability, and recyclability. The authors make a great contribution to investigate the formation mechanism, method universality and catalytic performance of the novel catalysts. Thereby I recommend acceptance of this work for publication after minor revisions have been made according the following comments:

Reply:

Thanks for your kind, careful and hard work to review our manuscript submitted to **Nature Communications** and giving the constructive suggestions to us which are very important for us to improve the quality of the previously submitted manuscript, and

also very importantly, for giving us a chance to revise this manuscript. According to your constructive suggestions, this manuscript has been carefully revised. We do hope after this revision our work can satisfy the high standard publication of **Nature Communications**.

MAJOR point:

Comment 1:

The scale bar in Figure 2 should be made more clearly, e.g., using the thicker white line.

Reply:

Thanks for your constructive suggestions. According to your suggestion, the scale bar in **Figure 2** has been readjusted and using the thicker white line.

Figure 2. FIB-AC-BF-TEM (A, B), FIB-AC-HAADF-TEM (C, D), FIB-AC-EDS Mapping (E), intra-mesopore size distribution (F), pore size distribution curve (G), Pd NPs size distribution (H) of Pd@IM-S-1.

Comment 2:

FIB technique is one of the audio-visual methods to confirm the confined structure. The

authors need to provide the measuring procedures in detail.

Reply:

Thanks for your constructive suggestions. According to your suggestion, we provide a more detailed measurement process, as detailed below.

Original:

The FIB specimens were obtained using the FEI Helios Nanolab 600i and working at 500 V–30 kV. First, the powder samples are attached to a carbon film, and then deposited a Pt protection layer on the surface of samples. Subsequently, position cutting at both ends of the Pt protection layer and welding manipulator on top of the thin sample layer. Next, take out the thin sample layer and welded it to the bracket, while remove the manipulator needle. Finally, the FIB specimens were obtained through the ion beam to thinning the thin sample layer and the simple flowchart are presented in **Figure S8**. Hereafter, the aberration-corrected TEM (AC-TEM) was applied to characterize FIB specimens via the FEI titan themis working at 300 kV equipped with an energy dispersive spectroscopy (EDS) detector.

Revised:

The FIB specimens were obtained using the FEI Helios Nanolab 600i from ZKKF (BEIJING) Science & Technology Co., Ltd. and working at 500 V–30 kV. Detailed steps are as follows:

- (1) **Sample pretreatment:** Attached non-conductive powder sample Pd@IM-S-1 to a carbon film.
- (2) **Selecting area and depositing Pt protective layer:** In the electron beam imaging mode of the double beam electron microscope, select the area of interest, and use the electron beam to deposit a layer of C or Pt with a thickness of about 0.5-1 microns, and then use the ion beam to deposit a layer of Pt with a thickness of 2-3 microns to protect the surface structure of the sample.
- (3) **Proposal and transfer:** Gradually dig out V-shaped grooves on both sides and cut off the root of the sample, extend the manipulator and weld the top of the sample, then cut the supporting parts on both sides of the sheet, take out the sample and transfer it to the support and weld it firmly. Finally, the sample was cut and separated from the tip of

the manipulator.

(4) **Thinning:** The FIB specimens were obtained through the ion beam to thinning the thin sample layer. The actual thinning parameters, including the acceleration voltage required to change the thickness and the tilt angle of the sample during the thinning process, mainly depend on the material system being studied. For general materials, the following parameters: use 30 KV to process the sample to ~1000 nm, then reduce the voltage to 16 KV, continue to thin to ~500 nm, and then reduce the voltage to 8 KV, continue to thin to ~200nm, and finally use 5 KV to thin to ~100 nm. If the sample needs to be characterized with high resolution, it is also necessary to reduce the voltage to 2 KV and gradually reduce the thickness of the sample to ~50 nm. After thinning to the target thickness, the sample is cleaned with a low voltage of 1 KV to remove the amorphous layer introduced by the ion beam.

The simple flowchart was presented in **Figure S8**. Hereafter, the aberration-corrected TEM (AC-TEM) was applied to characterize FIB specimens via the FEI titan themis working at 300 kV equipped with an energy dispersive spectroscopy (EDS) detector.

Comment 3:

As a hierarchical porous material, its external specific surface area needs to be provided in Table S1.

Reply:

Thanks for your constructive suggestions. According to your suggestion, we are provided the external specific surface area of Pd@IM-S-1 and added to **Table S1**.

Thanks again!

Table S1. Physicochemical properties and Pd content of related catalysts are measured by N₂ sorption isotherms and ICP.

Samples	SBET (m ² g ⁻¹) ^a	S _{ext} (m ² g ⁻¹) ^b	V _{micro} (cm ³ g ⁻¹) ^c	D _{micro} (nm) ^e	V _{meso} (cm ³ g ⁻¹) ^d	D _{meso} (nm) ^d	Pd content (wt.%) ^e
Pd@IM-S-1	408	76	0.158	0.68	0.136	2.5 (20) ^d	1.83
Pd/S-1	424	-	0.141	0.59	-	-	1.84

Pd@SiO ₂	93	-	0.014	1.4	0.407	36.6	1.73
S-1	452	-	0.167	0.74	-	-	-

^a Calculated by BET method.

^b S_{ext} (external surface area) calculated using the t-plot method.

^c Determined by HK method.

^d Determined by BJH method.

^e Obtained from ICP.

- not provided.

Comment 4:

Generally, the kinetic test should be carried out before excluding the mass transportation limits. Thus, I suggest the authors provide a detailed kinetic calculation to confirm that their kinetic results were obtained after excluding the mass transportation limits.

Reply:

Thanks for your careful and hard work to review our manuscript and kindly giving constructive suggestions to us, which are very important for us to improve the quality of the previously submitted manuscript. According to your suggestion, we provided a detailed kinetic calculation to confirm it and the kinetic results were obtained after excluding the mass transportation limits. The detailed process is as follows:

The absence of mass transport resistances was checked by Weisz-Prater criterion (C_{WP}) for internal diffusion and Mears' criterion (C_M) for external diffusion.

$$C_{WP} = \frac{r_{obs} \rho_c R_p^2}{D_{eff} C_s} < 1$$

$$C_M = \frac{r_{obs} \rho_b R_p n}{k_c C_{Ab}} < 0.15$$

Weisz-Prater Criterion for Internal Diffusion

If $C_{WP} = \frac{r_{obs} \rho_c R_p^2}{D_{eff} C_s} < 1$, then internal mass transfer effects can be neglected.

Where r_{obs} = observed reaction rate, mol/kg_{cat}·s

R_p = catalyst particle radius, m

ρ_c = density of catalyst, kg/m³

D_{eff} = effective diffusivity, m²/s

C_s = gas concentration of A at the external surface of the catalyst, mol/m³

The results of internal mass transfer are presented in **Table S6** and **Table S7**.

Table S6. The parameters of Weisz-Prater criterion (C_{WP}) for internal diffusion of Pd@IM-S-1 and related catalysts for CH₄ oxidation process.

Samples	ρ_c (kg/m ³)	$C_s \times 10^2$ (mol/m ³)	$R_p \times 10^4$ (m)	$r_{obs} \times 10^4$ (mol/kg _{cat} ·s)	$D_{eff} \times 10^6$ (m ² /s)	C_{WP}	C_{WP} compare 1
Pd@IM-S-1	542	44.5	1.35	26	8.9	0.006	<1
Pd/S-1	567	44.5	1.35	13	8.9	0.003	<1
Pd@SiO ₂	683	44.5	1.35	3.8	8.9	0.001	<1

Table S7. The parameters of Weisz-Prater criterion (C_{WP}) for internal diffusion of Pd@IM-S-1 and related catalysts for C₃H₈ oxidation process.

Samples	ρ_c (kg/m ³)	$C_s \times 10^2$ (mol/m ³)	$R_p \times 10^4$ (m)	$r_{obs} \times 10^4$ (mol/kg _{cat} ·s)	$D_{eff} \times 10^6$ (m ² /s)	C_{WP}	C_{WP} compare 1
Pd@IM-S-1	542	8.9	1.35	3.7	8.9	0.005	<1
Pd/S-1	567	8.9	1.35	1.5	8.9	0.002	<1
Pd@SiO ₂	683	8.9	1.35	0.5	8.9	0.001	<1

Because C_{wp} of CH₄ and C₃H₈ over the Pd@IM-S-1 and related catalysts are both below 1, thus the internal mass transfer effects can be neglected.

Mears Criterion for External Diffusion

If $C_M = \frac{r_{obs} \rho_b R_p n}{k_c C_{Ab}} < 0.15$, then external mass transfer effects can be neglected.

Where r_{obs} = observed reaction rate, mol/kg_{cat}·s

n = reaction order

R_p = catalyst particle radius, m

ρ_c = density of catalyst, kg/m³

ρ_b = bulk density of catalyst bed, kg/m³

$$= (1-\Phi) \rho_c \text{ (\Phi=porosity)}$$

$$\approx \rho_c \approx \rho_{\text{cat}}$$

C_s = gas concentration of A at the external surface of the catalyst, mol/m³.

C_{Ab} = bulk gas concentration of A, mol/m³.

$$\approx C_s$$

k_c = external mass transfer coefficient, m/s

The results of external mass transfer are presented in **Table S8** and **Table S9**.

Table S8. The parameters of Mears' criterion (C_M) for external diffusion of Pd@IM-S-1 and related catalysts for CH₄ oxidation process.

Samples	ρ_b (kg/m ³)	$C_{Ab} \times 10^2$ (mol/m ³)	n	$R_p \times 10^4$ (m)	$r_{\text{obs}} \times 10^4$ (mol/kg _{cat} ·s)	k_c (m ² /s)	C_M	C_M compare 0.15
Pd@IM-S-1	542	44.5	2	1.35	26	0.071	0.012	<0.15
Pd/S-1	567	44.5	2	1.35	13	0.071	0.006	<0.15
Pd@SiO ₂	683	44.5	2	1.35	3.8	0.071	0.002	<0.15

Table S9. The parameters of Mears' criterion (C_M) for external diffusion of Pd@IM-S-1 and related catalysts for C₃H₈ oxidation process.

Samples	ρ_b (kg/m ³)	$C_{Ab} \times 10^2$ (mol/m ³)	n	$R_p \times 10^4$ (m)	$r_{\text{obs}} \times 10^4$ (mol/kg _{cat} ·s)	k_c (m ² /s)	C_M	C_M compare 0.15
---------	----------------------------------	---	-----	--------------------------	--	------------------------------	-------	--------------------------

Pd@IM-S-1	542	8.9	2	1.35	3.7	0.071	0.009	<0.15
Pd/S-1	567	8.9	2	1.35	1.5	0.071	0.004	<0.15
Pd@SiO ₂	683	8.9	2	1.35	0.5	0.071	0.001	<0.15

Because C_M of CH₄ and C₃H₈ over the Pd@IM-S-1 and related catalysts are both below 0.15, thus the external mass transfer effects can be neglected.

Based on the above results (TableS6-S9), the internal and external diffusion effects during the kinetic experiment could be neglected (Applied Catalysis B: Environmental, 2020, 272, 118858; ACS Catalysis, 2019, 9 1472-1481).

Comment 5:

It is well known that the Pd or PdO_x species have activities for the addressed reactions. While the results provided in this manuscript suggest that the Pd-PdO interface are the active sites. The authors need to provide some important literatures to support their results.

Reply:

Thanks for your constructive suggestions. According to current reports, fundamental structural studies reveal that the metal–oxide interface plays a significant role in enhancing catalytic activity owing to strong interfacial interaction (**J. Mater. Chem. A, 2019, 7, 12627**). The interface of Pd-PdO has been reported as one of the most catalytically active species in hydrocarbons oxidation (**ACS Catal. 2017, 7, 4372-4380; Appl. Catal. B: Environ. 2018, 236, 436-444; Angew. Chem. Int. Ed. 2020, 59, 18522-18526**). According to your suggestion, we provided some important literature to support our results.

44. Goodman, E. D., Dai, S., Yang, A.-C., Wrasman, C. J., Gallo, A., Bare, S. R., Hoffman, A. S., Jaramillo, T. F., Graham, G. W., Pan, X., Cargnello, M., *ACS Catal.* **7**, 4372-4380 (2017).

45. Xiong, H., Wiebenga, M. H., Carrillo, C., Gaudet, J. R., Pham, H. N., Kunwar, D., Oh, S. H., Qi, G., Kim, C. H., Datye, A. K., *Appl. Catal. B: Environ.*, **236**, 436-444 (2018).

46. Yang, J., Peng M, Ren, G., Qi, H., Zhou, X., Xu, J., Deng, F., Chen, Z., Zhang, J., Liu, K., Pan, X., Liu, W., Su, Y., Li, W., Qiao, B., Ma, D., Zhang, T., *Angew. Chem. Int. Ed.* **59**, 18522-18526 (2020).

Comment 6:

Some of the important Figures or Scheme in the Supporting Information need to be moved to the main text.

Reply:

Thanks for your constructive suggestions. According to your suggestion, we are combined the **Figure S11-S18** and the **original-Figure 3** into a **new-Figure 3** and moved to the main text.

Figure 3. TEM image of Pd@SiO₂ (A), TEM images of Pd@IM-S-1 crystallized 1 day (B) and 2 days (C) at 80 °C; TEM images of Pd@IM-S-1 crystallized 1 h (D), 2 h (E),

4 h (F), 12 h (G), and 24 h (H) at 120 °C; XRD patterns of Pd@IM-S-1 crystallized between 80 to 120 °C at different crystallization time (I); proposed formation process (J) of intra-mesopore over Pd@IM-S-1.

REVIEWER COMMENTS

Reviewer #1 (Remarks to the Author):

Review of "In situ intra-crystalline mesoporous zeolite encapsulation-derived thermally roust metal nanocatalyst in deep oxidation" by H. Peng et al.

The manuscript of H. Peng et al. reports on the synthesis of Pd-based catalysts for the oxidation of light alkane (methane, propane) to CO₂ and H₂O, based on encapsulated Pd nanoparticles in zeolite crystals (silicalite-1) with tuned mesoporosity for better mass transport and Pd thermal stabilization. While the work reported is scientifically sound and the proposed synthesis strategy is interesting for catalysts design, the work lacks of new fundamental insights and does not meet the novelty required for publication in *Nature Communications*.

Pd thermal stabilization by encapsulation is a well-established strategy in heterogeneous catalysis (e.g. *Science* **2012**, 337, 713; *Nat. Commun* **2019**, 10, 1611), leading to high activity at low temperature in methane oxidation, comparable to those reported herein. Control of zeolite porosity aimed at metal encapsulation and stabilization is also a well-established field (e.g. *Front Chem.* **2018**, 6, 550). The Xray absorption results and in-situ infrared spectroscopy reported herein also do not shed new light into the chemistry of Pd under deep oxidation of alkanes or into the reaction mechanism (e.g. *ACS Catal.* **2015**, 5, 4, 2481–2489; *Nat. Commun* **2018**, 9, 2545; *Renew. Sust. Energ. Rev.*, **2020**, 119, 109589).

I therefore think the manuscript is not suitable for publication in *Nature Communication* and might be suitable for publication in another journal, such as *ACS Catalysis*. Some comments follow:

1. I would encourage the authors to proof-read the manuscript before resubmission: there is a typo even in the title ("roust" instead of "robust"). The title is also very obscure and should be rephrased (e.g. specifying "deep oxidation of alkanes", removing "in situ" from the title, etc.)
2. I suggest the authors to make the writings in all the figures much bigger, since they are very hard to read when printed on paper.
3. Figure 1 is not very clear, and is more fit for a table of content than a synthesis scheme.
4. Figure 3 is also not very dense with information. I would suggest to couple it with some evidence to support the proposed mechanism of mesopores formation.

Reviewer #2 (Remarks to the Author):

The authors have properly revised their manuscript according to the Reviewers' comments and it is now acceptable for publication.

Response to Reviewers' Comments

Ms. Ref. No.: NCOMMS-21-02997A

Title: Intra-crystalline mesoporous zeolite encapsulation-derived thermally robust metal nanocatalyst in deep oxidation of light alkanes

Reviewers' comments:

Reviewer #1 (Remarks to the Author):

Review of "Intra-crystalline mesoporous zeolite encapsulation-derived thermally robust metal nanocatalyst in deep oxidation of light alkanes" by Honggen Peng et al.

The manuscript of Honggen Peng et al. reports on encapsulated Pd nanoparticles in zeolite crystals for the catalytic oxidation of light alkanes (methane, propane) to CO₂ and H₂O. In the first review round, I express doubts about the novelty of the insights presented in the manuscript. I acknowledge the authors for taking the time to answer my comments at length. Nonetheless, I remain of the opinion that the manuscript is not suitable for publication in Nature Communications, and suggest submission to another catalysis oriented journal. The reasons for my recommendation are reported below.

Reply:

Thanks for your careful and hard work to review our revised version and kindly giving constructive suggestions to us, which are very important for us to improve the quality of the previously revised manuscript. According to your suggestions, we do our best to improve our manuscript. All the responses to your constructive suggestions are summarized in the following. Mainly: (1) By combining the density functional theory (DFT) calculations and *in-situ* near ambient pressure X-ray photoelectron spectra (*in-situ* NAP-XPS), we verify that the presence of Pd-PdO interfaces is beneficial for the deep oxidation of light alkanes; (2) The Pt, Rh, and Ru nanoparticles were indeed confined within IM-S-1 zeolite shell with the mean sizes of 5.8 nm, 2.3 nm, 2.3 nm, respectively evidenced by the Tomogram-section TEM; (3) Pd@IM-S-1-800 sample shows superior thermal stability with relatively small particles (5.6 nm) and large specific surface area (385 m² g⁻¹) after high temperature treatment (800 °C) verified by the Tomogram-section TEM and BET.

Specifically, we mentioned that the Pd–PdO interfaces, as new active sites, play a

critical role in the deep oxidation of light alkanes. In order to verify this conjecture, the density functional theory (DFT) calculations are performed. As is well-known, the cleavage of the first C-H bond is the rate-determining step in the deep oxidation of light alkanes. The existence of oxygen vacancies is conducive to the activation of gaseous oxygen molecules to generate reactive oxygen species, which contributes to the cleavage of C-H bonds. Therefore, the oxygen vacancy formation energy (E_v) of three different models including PdO (101), Pd (111) and Pd-PdO interface has been investigated. Taking methane combustion as an example, meanwhile, we have also studied the activation energy (E_a) of the methane first C-H cleavage over these three different models. The details of the calculation method and related models are presented in the Supplementary Information (**Figure S23-S29**). As can be seen in **Figure S25-S29**, the Pd-PdO interfaces show the lowest oxygen vacancy formation energy ($E_v = 1.38$ eV) and lowest activation energy ($E_a = 0.52$ eV) of the methane first C-H cleavage among these models. The results indicated that oxygen molecules are more easily activated on Pd-PdO interfaces and beneficial to the first C-H cleavage of light alkanes.

Figure S23. The optimized structure of Pd (111), PdO (101) and Pd-PdO interface.

Figure S24. Oxygen dissociation process on Pd (111) slab.

Figure S25. Oxygen vacancy formation energy (E_v) with different structure models.

Figure S26. The activation energy (E_a) of the methane first C-H cleavage over the Pd (111) model.

Figure S27. The activation energy (E_a) of the methane first C-H cleavage over the PdO (101) model.

Figure S28. The activation energy (E_a) of the methane first C-H cleavage over the Pd-PdO interface model.

Figure S29. The possible reaction pathways of methane first C-H cleavage over these different structure models.

To further investigate the changes of chemical valence of Pd species during the oxidation process, **in situ near ambient pressure X-ray photoelectron spectroscopy (in situ NAP-XPS)** was also performed (**Figure 6**). Pd species has two different states (Pd^0 and Pd^{2+}) under vacuum condition indicating some PdO species are still unreduced. The valence of Pd species remains unchanged at room temperature when the C_3H_8 and O_2 mixed gas ($\text{C}_3\text{H}_8:\text{O}_2 = 1:5$) was introduced. With the reaction temperature increasing to 225 °C, a new peak was detected around 346.7 eV in the $\text{Pd}3d_{3/2}$, which can be attributed to Pd^{4+} species (*Nat. Catal.* 2021, 4, 469–478; *Chem. Rev.* 2019, 119, 6822-6905). The peak intensity of Pd^{2+} and Pd^{4+} species gradually increased with the decrease of the Pd^0 species when the reaction temperature further increases, which indicated that some of the reduced metallic Pd species are oxidized to PdO_x species during the oxidation process.

Many other experiments were also performed to improve the quality of this manuscript according your constructive suggestions. We do hope after this revision, our manuscript can meet the publication standard of **Nature Communications**.
Thanks!

Figure 6. Pd speciation followed by the *in situ* NAP-XPS. Pd 3d core-line spectra as a function of reaction conditions for Pd@IM-S-1 at different temperature (from room temperature to 280 °C). The total pressure in the NAP cell was fixed at 1 mbar ($C_3H_8:O_2 = 1:5$).

MAJOR point:

Comment 1:

“a one-pot *in situ* strategy to fabricate single-crystal zeolites with intra-crystalline mesopores (intra-mesopores) and confine the active metal NPs simultaneously.”

However, the synthesis is not one-pot, as it requires the synthesis of Pd@SiO₂ first and then the “dry-gel conversion method” (see Scheme S1, notably it has two steps).

Reply:

Thanks for your careful and hard work to review our manuscript. We do agree with you that the synthesis of Pd@IM-S-1 has two steps, as it requires the synthesis of Pd@SiO₂ first and then the “dry-gel conversion of the Pd@SiO₂ to Pd@IM-S-1”. Due to the second step just adding the tetrapropylammonium hydroxide (TPAOH) as the microporous structure directing agent, like a one pot tandem reaction, thereby we use the “one-pot two-step” to illustrate the synthesis process. To avoid this misunderstanding, the “one-pot two-step method” was changed to “facile two-step method” in the revised manuscript. Thanks again!

Comment 2:

“the facile method exhibits versatility” and “Pt, Rh and Ru NPs are all confined in the IM-S-1 zeolite shell”. **However, it is not clear if Pt, Rh and Ru are in the pores: in Figure S14 Pt seems to be on the outside only, even more so for Rh in Figure S16.**

Reply:

Thanks for your careful and hard work to review our manuscript and pointing this out. As you mentioned, the TEM images are really not clear to confirm whether the Pt, Rh, Ru are confined within the IM-S-1 zeolite shell. Thus, the **Tomogram-section TEM** was performed to further prove the Pt, Rh, Ru are confined within IM-S-1 zeolite shell. As shown in **Figure S15, S18, S21**, most of the Pt, Rh, Ru are indeed confined within IM-S-1 zeolite shell and the mean size of Pt, Rh, Ru NPs are 5.8, 2.3 and 2.3 nm, respectively. Thanks again!

Figure S15. Tomogram-section TEM images (A, B, C) and Pt NPs size distribution (D) of 1%-Pt@IM-S-1.

Figure S18. Tomogram-section TEM (A, C) and HAADF-STEM (B) images, and Rh NPs size distribution (D) of 1%-Rh@IM-S-1.

Figure S21. Tomogram-section TEM images (A, B, C) and Ru NPs size distribution (D) of 1%-Ru@IM-S-1.

Comment 3:

The improved activity and thermal stability of the new catalyst. **Figure 5 B and E: the difference in activity of the fresh Pd@IM-S-1 and Pd/S-1 is minimal as can be observed by the T₁₀, T₅₀ and T₉₀. After aging at 800 °C, Pd@IM-S-1-800 and Pd/S-1-800 are also very similar (Figure 6A). Since Pd/S-1 is much easier to synthesize, the new approach does not seem very promising.**

Reply:

Pd@IM-S-1 and Pd/S-1 samples have clear differences in the activity for and propane oxidation (**Table R1**). For example, compared with Pd/S-1, T₅₀ of Pd@IM-S-1 shifted to the lower temperature (280 vs 261 °C for methane oxidation and 270 vs 260 for propane oxidation). These results reveal that Pd@IM-S-1 has better low-temperature methane and propane oxidation activity compared with Pd/S-1. After aging at 800 °C, the T₅₀ and T₉₀ over Pd@IM-S-1-800 (290, 324 °C) are still lower than Pd/S-1-800 (303, 334 °C) for propane oxidation, which also demonstrated that Pd@IM-S-1-800

possesses better propane oxidation activity than Pd/S-1-800. In addition, the reaction rates and turnover frequency for methane and propane oxidation, calculated at 265 °C for Pd@IM-S-1, are two times higher than those over Pd/S-1 and seven times those over Pd@SiO₂. Thus, Pd@IM-S-1 displays superior deep oxidation performance compared with Pd/S-1 and Pd@SiO₂, indicating that Pd@IM-S-1 is a promising catalyst for deep oxidation of light alkanes. Thanks again!

Table R1. Catalytic activities and kinetic catalytic performances of Pd@IM-S-1 and Pd/S-1.

Samples	T ₅₀ (°C)		T ₉₀ (°C)		Rate (mol g _{cat.} ⁻¹ s ⁻¹)		TOF ^{total} (10 ³ s ⁻¹)	
	CH ₄	C ₃ H ₈	CH ₄	C ₃ H ₈	CH ₄ (265 °C)	C ₃ H ₈ (265 °C)	CH ₄ (265 °C)	C ₃ H ₈ (265 °C)
Pd@IM-S-1	261	260	318	284	2.6×10 ⁻⁶	3.7×10 ⁻⁷	14.9	2.14
Pd/S-1	280	270	325	297	1.3×10 ⁻⁶	1.5×10 ⁻⁷	7.4	0.84

Comment 4:

Moreover, the effect of water (Figure 6C) shows a weird baseline for three lines at low temperature, and the regeneration in Figure 6F should be reported also for the Pd/S-1 catalyst for comparison.

Reply:

Thanks for your constructive suggestions. Just as you mentioned, the effect of water shows a weird baseline for three lines at low temperature (**Figure 7C**). This experiment was repeated three times and exhibited the similar trend. It is indeed like that. The difference at low temperature is due to the worse water resistance of Pd/S-1 than Pd@IM-S-1. The better water resistance of Pd@IM-S-1 is probably owing to the hydrophobic and guarding effect of the zeolite shell, which prohibits the direct exposure of the active components to water vapor.

Furthermore, the regeneration test of Pd/S-1 has been further investigated as shown in **Figure S34**. It can be seen that the propane oxidation activity was sharply declined over Pd/S-1-5th-run after the fifth cycling test and the activity over Pd/S-1-restore

could not be regenerated. However, the propane oxidation activity over Pd@IM-S-1-restore was almost completely regenerated as shown in **Figure 7F**. These results demonstrated that the recyclability of Pd@IM-S-1 was better than that of Pd/S-1. Thanks again!

Figure S34. Regeneration test of Pd/S-1 in the oxidation of propane.

Comment 5:

Furthermore, a much more detailed characterization (e.g. TEM, BET) should be performed on the high temperature aged catalysts, as resistance to sintering is one of the main issues that the authors are trying to tackle. At the moment, the thermal stabilization obtained by introducing the mesopores is not convincing. On a conceptual level, it is also not clear how encapsulation can be assured in mesopores: why should Pd(O) NPs not grow until they fill a certain mesoporous structure, and what is stopping them to migrate if they are not encapsulated in smaller pores?

Reply:

Thanks for your constructive suggestions. According to your suggestion, the Tomogram-section TEM and BET of Pd@IM-S-1-800 was performed. As can be seen

in **Figure S33**, the Pd species still maintains relatively small particles (5.6 nm) even though the Pd species grows to relatively larger particles (from 2.4 nm and 4.0 nm to 5.6 nm) after high temperature treatment. Moreover, most of the Pd NPs still were confined within zeolite shell and homogeneously distributed over all of the zeolite. In addition, the results of BET test of Pd@IM-S-1-800 are present in **Table S1**. It can be seen that the specific surface area of Pd@IM-S-1-800 was slightly decreased (from 408 m² g⁻¹ decreased to 385 m² g⁻¹) after high temperature treatment. The Tomogram-section TEM and BET results demonstrates that Pd@IM-S-1 has good thermal stability due to the confinement effect of zeolite shell. During high temperature treatment process, Pd species with the smaller NPs than the channel sizes of zeolite will migrate and grow into larger particles. When the Pd NPs size was larger than the channel sizes of zeolite, it will prevent the further growth of Pd species. In our future work, the active metal oxides (MnO_x, SnO₂, TiO₂, etc.) will be introduced to modify the metal@IM-S-1 catalyst to further stabilize the metal core through the strong metal-metal oxides interaction. Thanks again!

Figure S33. Tomogram-section TEM images (A, B, C) and Pd NPs size distribution (D) of Pd@IM-S-1-800.

Table S1. Physicochemical properties and Pd content of related catalysts are measured by N₂ sorption isotherms and ICP.

Samples	S _{BET} (m ² g ⁻¹) ^a	S _{ext} (m ² g ⁻¹) ^b	V _{micro} (cm ³ g ⁻¹) ^c	D _{micro} (nm) ^c	V _{meso} (cm ³ g ⁻¹) ^d	D _{meso} (nm) ^d	Pd content (wt.%) ^e
Pd@IM-S-1	408	76	0.158	0.68	0.136	2.5 (20) ^d	1.83
Pd@IM-S-1-800	385	66	0.129	0.58	0.179	2.4 (4.0)	-
Pd/S-1	424	-	0.141	0.59	-	-	1.84
Pd@SiO ₂	93	-	0.014	1.4	0.407	36.6	1.73
S-1	452	-	0.167	0.74	-	-	-

^a Calculated by BET method.

^b S_{ext} (external surface area) calculated using the t-plot method.

^c Determined by HK method.

^d Determined by BJH method

^e Obtained from ICP.

- Not provided.

Comment 6:

“the catalytic performance of Pd@IM-S-1 in our work was better than that of Pd@CeO₂/Al₂O₃ and Pd/NA-Al₂O₃, and our work shows a unique advantage.” **The authors fail to acknowledge the very different WHSV used in the studies (e.g. 36000 vs 200000 mL g⁻¹ h⁻¹), and the pretreatment temperatures used (550 vs 800 °C). When aged at 800 °C, the catalyst was already much worse in terms of performance.** I strongly suggest the authors employ more relevant conditions for their study (e.g. 200000 mL g⁻¹ h⁻¹ for methane oxidation).

Reply:

Thanks for your constructive suggestions. According to your suggestion, the methane oxidation activity over Pd@IM-S-1-800 at higher WHSV (200000 mL g_{cat.}⁻¹ h⁻¹) was performed as shown in **Figure R2**. It can be seen that the T₅₀ and T₉₀ over Pd@IM-S-1-800 are 337 and 385 °C, respectively. For Pd@CeO₂/Al₂O₃ (*Science*,

2012, 337, 713), the T_{50} and T_{90} are ~ 325 and ~ 350 °C, respectively. Therefore, Pd@IM-S-1-800 and Pd@CeO₂/Al₂O₃ have very similar catalytic performance. However, the methane content of their work is only 0.5%, while our work is as high as 1.0%. For Pd/NA-Al₂O₃, the Pd content is as high as 5 wt% and the WHSV is as low as 15000 ml g⁻¹ h⁻¹. Thus, it is very difficult to compare the catalytic performances over different catalysts in various literature works equally. Thanks again!

Figure R2. Deep oxidation performances of methane over Pd@IM-S-1-800 at high WHSV. Conditions: 1% CH₄, 21% O₂, N₂ balance, and the WHSV was 200000 mL g_{cat.}⁻¹ h⁻¹.

Comment 7:

“promoting the mass-transfer efficiency of reactants and products.” **But the authors show (response to Referee 2, comment 4, and SI added sections) that mass transfer was not a problem in any of the studied catalysts. Can the authors clarify and support their claims with evidence?**

Reply:

Thanks for your careful and hard work to review our manuscript. As we all know, the diffusion limitation has always existed in heterogeneous catalysts and greatly affect the activity of the catalysts (*Chem. Rev.* 2020, 120, 20, 11194–11294; *Chem. Soc. Rev.*, 2016, 45, 3353-3376; *Chem. Soc. Rev.*, 2016, 45, 3313-3330; *Chem Eng. J.*, 2020, 385, 123800). There are two main ways to solve this problem, reducing the particle size and introducing multiple pores in the catalysts. The introducing of mesoporous on zeolite catalysts is a commonly method to reduce the diffusion limitation, thereby improving the mass transfer efficiency of reactants and products. There have been many related studies (*Angew. Chem. Int. Ed.* 2019, 59, 3455-3459; *J. Am. Chem. Soc.* 2019, 141, 3772-3776; *J. Am. Chem. Soc.* 2014, 136, 2503-2510) in this area, even though the provided review literature (*Front Chem.* 2018, 6, 550) in the first review round has also mentioned the introducing of mesopores can improve the mass transfer efficiency.

Additionally, the prerequisites for intrinsic kinetic testing must exclude internal and external diffusion limitations firstly, then the accurate data can be obtained. The common method is to eliminate the influence of internal and external diffusion by increasing the weight hourly space velocity (WHSV) to exclude the external diffusion at low conversion (usually < 15%) to perform the intrinsic kinetic test. Thus, we increase the WHSV from 36000 mL g_{cat.}⁻¹ h⁻¹ to 180000 mL g_{cat.}⁻¹ h⁻¹ to investigate the intrinsic kinetic in our manuscript. According to the suggestion of the second reviewer, we need to confirm whether the internal and external diffusion have been eliminated in intrinsic kinetic test. Therefore, the **Weisz-Prater criterion (C_{WP})** and **Mears' criterion (C_M)** were applied to verify whether the internal and external diffusion have been eliminated. **Universally, if C_{WP} < 1 and C_M < 0.15, the internal and external diffusion effects could be neglected.** Based on the calculation results (**Table S6-S9**), we can see that the internal and external diffusion effects could be neglected. We are sorry for our unprecise description of “mass transfer was not a problem in any of the studied catalysts”. **The conclusion of “the external mass transfer effects can be neglected” was only based on the kinetic test conditions.** To avoid this

misunderstanding, we have made a clearer description. Thanks again!

Comment 8:

In the rebuttal point-by-point, the authors mention that their XAS experiments were done ex-situ, while literature was done in-situ. **This is not just a difference, ex-situ experiments are less indicative of the actual state of the catalyst, and therefore not a state-of-the-art characterization.** This is particularly relevant for PdO, which can decompose to Pd and reoxidize depending on temperature, and thus will evolve even during cooling down from reaction temperature.

Reply:

Thanks for your careful and hard work to review our manuscript. The ex-situ XRD, Raman and XPS characterizations showed that the chemical valence of Pd species was changed from Pd⁰ to Pd²⁺ during the oxidation process of propane. In order to further study the changes of chemical valence of Pd species during the oxidation process, the **in situ near ambient pressure X-ray photoelectron spectroscopy (in situ NAP-XPS)** was performed as shown in **Figure 6**. Pd species has two different states (Pd⁰ and Pd²⁺) under vacuum condition indicating some PdO species are still unreduced. The valence of Pd species remains unchanged at room temperature when the C₃H₈ and O₂ mixed gas (C₃H₈:O₂ = 1:5) was introduced. As the reaction temperature increased to 225 °C, a new peak was detected around 346.7 eV in the Pd3d_{3/2}, which can be attributed to Pd⁴⁺ species (*Nat. Catal.* 2021, 4, 469–478; *Chem. Rev.* 2019, 119, 6822-6905). The peak intensity of Pd²⁺ and Pd⁴⁺ species gradually increased with the decrease of the Pd⁰ species when the reaction temperature further increases, which indicated that some of the reduced metallic Pd species are oxidized to PdO_x species. Therefore, the ex-situ XRD, Raman, XPS and in situ NAP-XPS results all indicate that the changes of chemical valence of Pd species are from Pd⁰ to high-valence Pd²⁺ and Pd⁴⁺ during the oxidation process of propane. Thanks again!

Figure 6. Pd speciation followed by in situ NAP-XPS. Pd 3d core-line spectra as a function of reaction conditions for Pd@IM-S-1 at different temperature (from room temperature to 280 °C). The total pressure in the NAP cell was fixed at 1 mbar ($C_3H_8:O_2 = 1:5$).

Comment 9:

Moreover, in my opinion the propane reaction mechanism section is not well-fitted in the overall story.

Reply:

Thanks for your careful and hard work to review our manuscript. It is very difficult to design a high efficiency catalyst used for pollutant elimination, because of the complicated reaction process, particularly, catalytic oxidation of complex pollutant molecules. Nowadays, researchers are paying more and more attention to the study of the mechanism of heterogeneous catalytic reactions, because it is beneficial for design

of high-performance catalysts. A lot of useful information can be obtained through the study of reaction mechanism, such as the reaction path of the pollutant molecules on the catalysts and the deactivation mechanism of the catalyst itself, etc. Therefore, the investigation of reaction mechanism is significantly and meaningful, which is helpful to provide advantageous reference basis for the design of other high-performance catalysts. **We do agree with you. According to your suggestion, to fit well in the overall story, we have weakened the discussion of the propane reaction mechanism and moved the *in situ* DRIFTS analysis to the Supplementary Information. Thanks!**

Reviewer #2 (Remarks to the Author)

The authors have properly revised their manuscript according to the Reviewers' comments and it is now acceptable for publication.

Reply:

Thanks for your constructive suggestions again!

REVIEWERS' COMMENTS

Reviewer #1 (Remarks to the Author):

I sincerely thank the authors for answering all of my comments.

I think the manuscript has much improved and I can recommend publication in the present form.